# Progressively Compressed Autoencoder for Self-supervised Representation Learning

**Jin Li**[1,†]   **Yaoming Wang**[1,†]   **Xiaopeng Zhang**[2]   **Yabo Chen**[1]
**Dongsheng Jiang**[2]   **Wenrui Dai**[1]   **Chenglin Li**[1]   **Hongkai Xiong**[1]   **Qi Tian**[2*]
[1]Shanghai Jiao Tong University   [2]Huawei Cloud
{deserve_lj, wang_yaoming, chenyabo, daiwenrui, lcl1985,
xionghongkai}@sjtu.edu.cn; zxphistory@gmail.com,
dongsheng jiang@outlook.com, tian.qi1@huawei.com

## Abstract

As a typical self-supervised learning strategy, Masked Image Modeling (MIM) is driven by recovering all masked patches from visible ones. However, patches from the same image are highly correlated and it is redundant to reconstruct all the masked patches. We find that this redundancy is neglected by existing MIM based methods and causes non-negligible overheads in computation that do not necessarily benefit self-supervised representation. In this paper, we present a novel approach named **PCAE**, short for **P**rogressively **C**ompressed **A**uto**E**ncoder, to address the redundant reconstruction issue by progressively compacting tokens and only retaining necessary information for forward propagation and reconstruction. In particular, we identify those redundant tokens in an image via a simple yet effective similarity metric between each token with the mean of the token sequence. Those redundant tokens that other ones can probably represent are progressively dropped accordingly during the forward propagation, and importantly, we only focus on reconstructing these retained tokens. As a result, we are able to achieve a better trade-off between performance and efficiency for pre-training. Besides, benefitting from the flexible strategy, PCAE can be also directly employed for downstream fine-tuning tasks and enable scalable deployment. Experiments show that PCAE achieves comparable performance to MAE with only 1/8 GPU days. The code is available at `https://github.com/caddyless/PCAE/`

## 1 Introduction

Contrastive learning has witnessed great progress and even outperforms its supervised counterpart in downstream tasks (Chen et al., 2020b; Caron et al., 2021; Grill et al., 2020). However, contrastive-based methods usually take more epochs and computational overheads for considerable performance. Recently, Masked Image Modeling (MIM) becomes a popular topic for its scalability as well as promising performance (Bao et al., 2022; He et al., 2022; Dong et al., 2021; Chen et al., 2022; Xie et al., 2022), especially for MAE (He et al., 2022) which significantly accelerates training via only operating on 25% visible patches in the encoder.

MIM methods learn representations by recovering masked regions of input images from visible ones. These masked regions are represented with learnable mask tokens in pre-training. Since MIM methods usually apply high mask ratios on the input image, the number of mask tokens is considerable. As a result, these mask tokens take large computational overheads during training while contributing little information. For example, the number of mask tokens is three times the visible tokens in the decoder of MAE, resulting in the overhead of the decoder (5.28G Flops) exceeding that of the encoder (4.3G Flops) for a typical ViT Base, even though the decoder is relatively lightweight. Another type of MIM methods, i.e., SimMIM (Xie et al., 2022; Bao et al., 2022) alike methods, retain masked tokens in the complicated encoder, causing a more serious computational burden. However, due to the spatial redundancy of images, the masked regions are highly correlated and redundant,

---

*: Corresponding author. †: Equal contribution. This work was done when Jin Li and Yaoming Wang worked as interns at Huawei Cloud.

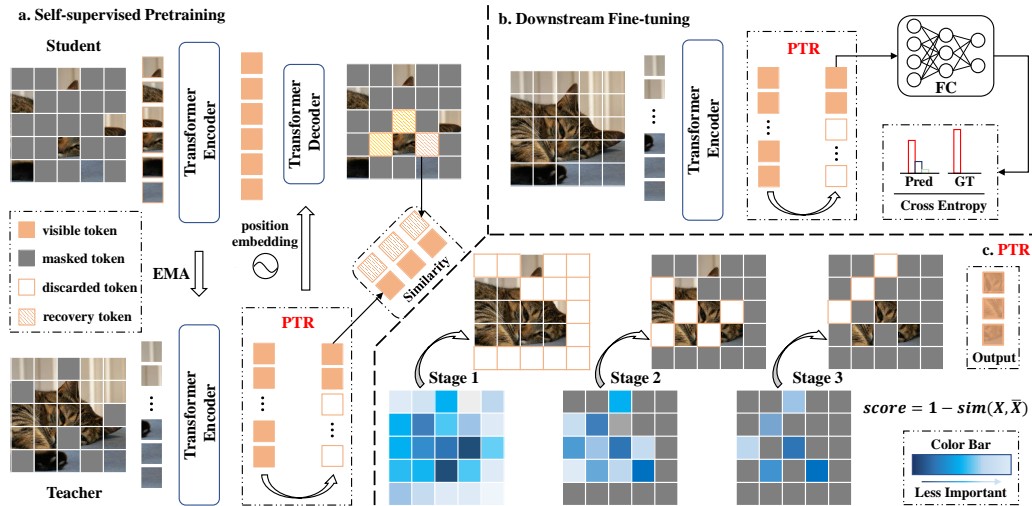

Figure 1: The overall framework of PCAE. (a) The pre-training framework consists of the student, the teacher (EMA model), and the decoder module. The student operates on visible patches while the teacher operates only on masked ones. The teacher progressively discards redundant tokens and ultimately outputs much fewer tokens for reconstruction. The decoder reconstructs the output of the teacher according to the output of the student. (b) PCAE for downstream fine-tuning. (c) Progressive Token Reduction (PTR) module. PTR can be applied in the teacher for pre-training or the backbone for downstream fine-tuning.

and thus it is unnecessary to recover all of them. Unfortunately, previous MIM methods neglect this and recover all masked regions through plenty of mask tokens, which causes non-negligible overheads in computation that do not necessarily benefit self-supervised representation. This observation motivates us to relax MIM pre-training by reducing the redundancy in reconstruction targets.

In this paper, we present a novel approach named Progressively Compressed Auto-Encoder (PCAE) to reduce the redundancy in reconstruction targets. The core idea is, instead of recovering all masked patches, we identify and neglect those redundant ones and only reconstruct the representative targets. However, naively removing part of the reconstruction targets suffers serious performance degradation (please refer to Table 9 for more details) as some important information is also discarded. Therefore, we propose to mitigate information loss by exploiting the self-attention mechanism of the vision transformer to spread information from the discarded patches to the retained ones. Specifically, we employ the momentum encoder in pre-training and discard tokens produced by the momentum encoder, rather than the original reconstruction targets.

To further alleviate the information loss, we propose a progressively discarding strategy where redundant tokens are progressively discarded at different layers of the momentum encoder (please refer to Table 3 for more details). The remained issue is how to identify the redundant tokens. Matrix decomposition precisely identifies the redundant component of the token sequence but exhibits prohibitive complexity in practice. As an alternative, we propose a simple yet effective criterion based on their similarity to the mean of the token sequence. Experimental results and visualizations demonstrate that this criterion is very effective and introduces negligible overhead.

Benefiting from the proposed progressive token reduction (PTR) strategy, PCAE reduces the reconstruction targets from 147 tokens to 18 tokens (take ViT Base as an example), which significantly improves the efficiency. Experiments over multiple benchmarks demonstrate the effectiveness of the proposed method. Specifically, PCAE is able to accelerate training 2.25 times compared with MAE (He et al., 2022) (739.7 img/s v.s. 328.4 img/s, ViT Base, 32 GB V100), while enjoying much faster converge speed (PCAE 300 epoch 83.6 vs MAE 1600 epoch 83.6) or higher performance (PCAE 800 epoch 83.9 vs MAE 1600 epoch 83.6). Besides, we extend the proposed strategy to the inference phase as well as downstream fine-tuning for classification tasks, and the results show this strategy still works well. Benefiting from the flexibility, we could provide models with different in-

ference speeds by adjusting the number of discarded tokens. Specifically, compared with the vanilla model, we accelerate 15% to 57% throughput while the performance drop is within 0.6%.

In summary, we list our contributions as follows:

- We revisit the paradigm of MIM and point out that the reconstruction task in MIM is redundant, which is ignored by previous methods.
- We propose a novel self-supervised learning approach, termed PCAE to accelerate the self-supervised pre-training without compromising performance.
- Our proposed PCAE can be adapted to downstream fine-tuning and achieves unified acceleration in pre-training and downstream tasks.

## 2 RELATED WORK

**Self-supervised learning.** Self-supervised learning constructs the learning target from the input signal and expects to generalize the obtained knowledge to downstream tasks. One taxonomy of self-supervised learning is to divide it into contrastive and generative (Liu et al., 2021). Contrastive methods (Chen et al., 2020a; Wu et al., 2018; He et al., 2020; Chen et al., 2020b; Caron et al., 2020; 2021; Zbontar et al., 2021; Li et al., 2021; Grill et al., 2020) have been widely studied and achieved great success on multiple downstream tasks including classification, object detection, and semantic segmentation (Chen et al., 2020b). Recent advanced generative methods are dominated by Masked Image Modeling (MIM) methods, which are driven by restoring masked patches from visible ones. MIM methods (Chen et al., 2022; He et al., 2022; Wei et al., 2022) exhibit superior performance to previous contrastive learning methods in downstream fine-tuning tasks based on the modern vision transformers. In this paper, we mainly focus on MIM methods.

Predicting the masked part of the image is a classic self-supervised task, and has been researched by previous works (Doersch et al., 2015; Pathak et al., 2016) based on the convolutional neural network. However, due to their limited performance, these methods are located in a non-mainstream position for a long time, especially after the advent of contrastive learning. Recently, the vision transformer (Dosovitskiy et al., 2021) bridges the architecture gap between vision and language, and more MIM methods (Bao et al., 2022; Dong et al., 2021; Xie et al., 2022; He et al., 2022; Wei et al., 2022; Chen et al., 2022) are proposed inspired by Masked Language Modeling. BEiT (Bao et al., 2022), PeCo (Dong et al., 2021), and CAE (Chen et al., 2022) rely on the pre-trained dVAE to produce the prediction target. MAE (He et al., 2022) and SimMIM (Xie et al., 2022) directly predict the raw pixel value, and MaskFeat (Wei et al., 2022) predicts the HOG (Dalal & Triggs, 2005) feature of masked patches. Despite predicting different kinds of targets, these methods reconstruct all masked patches at the output end, thus still suffering the redundant reconstruction task.

**Dynamic vision transformer.** Several methods (Rao et al., 2021; Meng et al., 2022; Yin et al., 2022; Liang et al., 2022) have been proposed to dynamically discard inattentive tokens for efficiency in vision transformers. However, these methods are only accessible for the supervised setting due to their complicated design and requirement on supervisory signals. Specifically, Dynamic ViT (Rao et al., 2021), A-ViT (Yin et al., 2022) and Ada-ViT (Meng et al., 2022) require to initialize from pre-trained models and only accelerates the inference. E-ViT (Liang et al., 2022) measures token importance via the attention to the class token but requires supervisory signals on class tokens. Zeng et al. (2022) proposes to merge patches that shared similar semantic concepts while keeping fine resolution for regions containing critical details. However, it relies on specific architectures and supervisory signals. In summary, although dynamic vision transformers have been explored in supervised learning, their application to self-supervised learning is still blank.

## 3 METHOD

This section is organized as follows: Initially, section 3.1 presents the overall framework and objective of PCAE. Section 3.2 elaborates on how the compact reconstruction targets are built including identifying redundant tokens and the progressively discarding strategy. Section 3.3 demonstrates the effectiveness of PCAE through illustrating the diversity of token sequence at different layers and visualizing the remaining tokens at each stage. Section 3.4 extends PCAE to downstream classification task so that the proposed strategy could improve efficiency across pre-training, fine-tuning, and final inference.

## 3.1 OVERVIEW

The framework of PCAE for pre-training is illustrated in Figure 1 (a), and the overall objective of PCAE is:

$$\max_{\theta,\phi,\omega} \mathbb{E}_{x\sim\mathcal{X}} \mathcal{M}\{\bar{f}[(1-M)\odot x;k], g_\phi\{h[f_\theta(M\odot x),\omega+p]\}\} \tag{1}$$

where $\odot$ is the element-wise product; $M$ is the random mask and $M\odot x$ denotes unmasked patches while $(1-M)\odot x$ denotes masked patches; $f_\theta$ is the encoder to be pre-trained while $\bar{f}$ is the exponential moving average model of $f_\theta$; $k$ is the ratio of tokens we expect to keep. $p$ is the positional embedding of tokens requested to reconstruct and $\omega$ is the expanded learnable mask token which has the same shape as $p$; $h(\cdot,\cdot)$ is the concatenation operation; $g_\phi$ is the decoder network; $\mathcal{M}(\cdot,\cdot)$ is the similarity measurement, and we instantiate it with cosine similarity here.

The EMA model $\bar{f}$ operates on masked patches and progressively discards redundant tokens from the token sequence, thereby ultimately outputting much fewer tokens. On the other end, the encoder process unmasked patches and outputs $f_\theta(M\odot x)$. Then, the decoder $g_\phi$ concatenates the output of the encoder and the position embedded mask tokens to reconstruct $\bar{f}((1-M)\odot x;k)$. Note that, as a reminiscent method, MAE (He et al., 2022) expects to reconstruct all patches, so the number of mask tokens is 147 (take ViT-B/16 as an example). In contrast, PCAE only reconstructs 18 tokens, and the computational overhead on the decoder could be largely reduced.

## 3.2 PROGRESSIVE TOKEN REDUCTION

In this section, we elaborate on the EMA $\bar{f}$ where we progressively discard redundant tokens. We discuss the progressively discarding strategy and redundant token identification in the following.

**Progressively discarding strategy.** As vision transformers are stacked by self-attention blocks, it is important to decide at which layer to discard tokens. Intuitively, for a given keep ratio, discarding tokens at once suffers serious information loss, and progressively discarding them at different layers could largely alleviate it by exploiting the information propagation capability of the self-attention mechanism. Formally, we decompose $\bar{f}$ into $\bar{f}_1\circ\bar{f}_2\circ\cdots\circ\bar{f}_L$ where $\bar{f}_i$ indicates the $i^{th}$ layer in $\bar{f}$. Then, for the output of $i^{th}$ layer $x_i\in\mathbb{R}^{N_i\times D}$ where $N_i$ indicates the number of tokens in the $i^th$ layer and $D$ indicates the number of dimensions, we have:

$$x_i = \begin{cases} T(\bar{f}_i(x_{i-1});k_i), & \text{if } i\in I \\ \bar{f}_i(x_{i-1}), & \text{if } i\notin I \end{cases} \tag{2}$$

where $I$ is the set of layers at which we expect to discard tokens; $T(\cdot;\cdot)$ is the redundant token identification function and we will discuss it later; $k_i$ is the ratio of tokens we expect to keep in $i^{th}$ layer. In practice, we apply the consistent drop ratio $k$ for all stages. Note that as we perform the token reduction in the momentum encoder, the gradient backpropagation is not taken into consideration. Thus, once a token is discarded, it will never be used in later layers, which accelerates the inference of the momentum encoder.

**Redundant token identification.** Matrix decomposition precisely identifies the redundant component in token sequence but exhibits prohibitive complexity in practice (slow down training five times in our implementation). As an alternative, we turn to identify redundant tokens according to their similarity to the average token. We are motivated by two reasons. First, removing tokens most similar to the average token has the smallest influence on the average of the new token sequence. Specifically, considering the average of the original token sequence $\bar{a}$, the average of remaining tokens $a_r$ and the average of discarded tokens $a_d$, it is obvious that $\|\bar{a}-a_r\|\propto\|\bar{a}-a_d\|$. Therefore, this strategy has minimal impact on the overall distribution of tokens. Second, according to the Discrete Fourier Transform (DFT), the average token corresponds to the low-frequency component of the image which usually presents as the background, and it is reasonable to remove them as they contribute little to visual perception. Formally, for a token sequence $x\in\mathbb{R}^{N_i\times D}$, the redundant token identification can be given as:

$$S(x) = rank(\mathcal{M}(x, \sum_{j}^{N_i} x^j/N_i)) \in \mathbb{R}^{N_i}, \tag{3}$$

$$T(x;k) = gather[x, S(x) < \lfloor N_i * k \rfloor] \in \mathbb{R}^{\lfloor N_i * k \rfloor \times D}$$

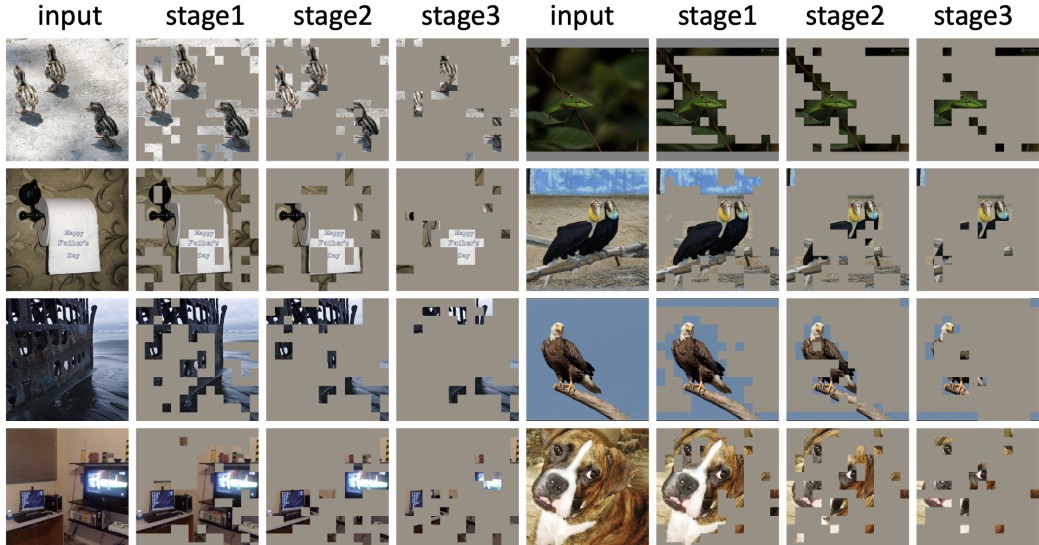

Figure 2: The visualization of the proposed stage-wise discarding strategy. Each row consists of two examples. For each example, we show (1) the input image and (2)-(4) images where corresponding patches are masked after stage 1, stage 2, and stage 3, respectively.

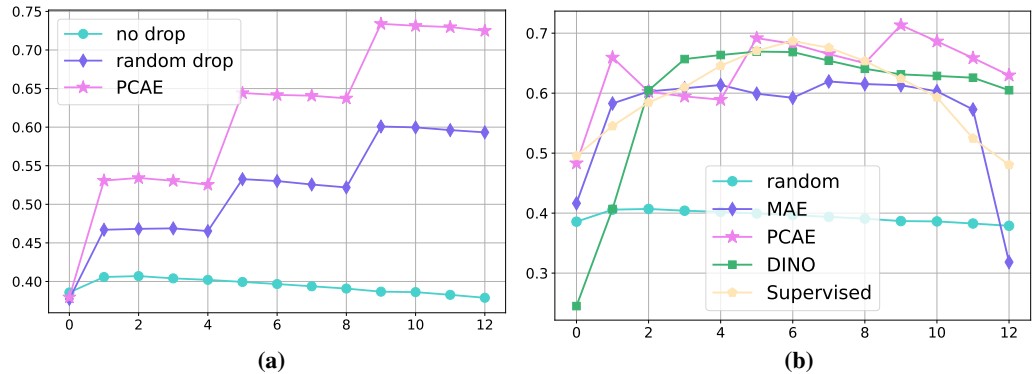

Figure 3: We measure the diversity of token sequences during the forward process. The vertical axis indicates the value of the nuclear norm, and the horizontal axis indicates the index of the layer (0 indicates the input token sequences). (a) We run the randomly initialized ViT-Base network on ImageNet validation set with three different settings to show their differences in token diversity. (b) We compare the curve which describes the relationship between the layer number and the nuclear norm between models initialized by different methods.

where $\mathcal{M}$ is the similarity measurement and instantiated with cosine similarity; $rank(\cdot)$ denotes the rank function in ascending order; $S(\cdot)$ is the rank of tokens; $gather(\cdot, \cdot)$ indicates the gather function where the rows of $x$ satisfy the right side condition are collected. k is the ratio of tokens we expect to keep. Eq. 3 shows that we determine whether a token is redundant according to its relative similarity to the average token. Tokens that are more similar to the average token than other k tokens are regarded as redundant tokens, and vice versa. The output of $\bar{f}$ could be given by combining Eq. 3 and Eq. 2.

## 3.3 ANALYZE

**Diversity enhancement by PCAE.** We run randomly initialized ViT-B networks on ImageNet validation set under different settings and measure the average diversity at each layer (0 indicates the input token sequences) with the nuclear norm (more details please refer to the supplementary) over the whole ImageNet validation set. As illustrated in Figure 3a, the proposed strategy effectively enhances the diversity of token sequence compared with cases where we do not discard tokens or randomly discard tokens.

Table 1: Image classification results on ImageNet dataset with top-1 accuracy. All methods are run under 1024 batch size for fair comparisons. $^\dagger$ indicates this method needs pre-trained dVAE.

| Method | Epochs | Forwards | GPU Days | Accuracy |
|---|---|---|---|---|
| Train from Scratch | 300 | 1 | - | 81.8 |
| MoCo v3 | 300 | 2 | - | 83.2 |
| DINO | 400 | 12 | 122.5 | 83.3 |
| BEiT$^\dagger$ | 300 | 1 | - | 83.0 |
| CAE$^\dagger$ | 300 | 2 | - | 83.6 |
| SimMIM | 800 | 1 | 54.1 | **83.8** |
| MAE | 300 | 1 | 15.4 | 82.9 |
| MAE | 1600 | 1 | 82.1 | 83.6 |
| PCAE | 300 | 2 | 10.1 | 83.6 |
| PCAE | 800 | 2 | 26.9 | **83.9** |

In addition, we also apply this setting to pre-trained checkpoints to reveal the differences between pre-training methods. As illustrated in Figure 3b, we find that all pre-training methods no matter whether supervised or unsupervised enhance the diversity of token sequence at intermediate layers. We hypothesize that meaningful pre-training makes the network learn to focus on important things and neglect redundant components in input signals. Another interesting finding is that MAE suffers low diversity at both the input end (index 0) and the output end (index 12). We hypothesize that it is caused by the symmetry of its input and output. In contrast, PCAE overcome the low diversity problem at its output end.

**Visualization of PCAE.** We visualize patches corresponding to the remaining tokens after each stage, and the results are presented in Figure 2. We find that this visualization is exactly in accord with the frequency perspective we discussed above. Referring to the second example in the third row, PCAE filters most of the sky background and eventually keep the detail-rich patches such as the beak and claws. It also holds for other images, such as the three chicks in the first example in the first row, the snake's head in the second example in the first row, and the text in the first example in the second row. All these details are well preserved.

We also find that PCAE not only focuses on the salient object, but also retains a variety of objects within the image. As illustrated in Figure 2, tokens kept by PCAE are diverse and distributed throughout the image, which preserves the diversity within the image. We attribute it to the stage-wise discard strategy. Referring to the second example in the third row, the first two stages discard most patches belonging to the sky. Therefore, in the third stage, patches belonging to the sky are no longer redundant, thereby being preserved eventually. The stage-wise strategy helps preserve the diversity within images.

### 3.4 DOWNSTREAM FINE-TUNING

As the proposed strategy is easy to implement and does not require any extra parameters, it can be seamlessly incorporated into inference as well as downstream fine-tuning. Specifically, we transfer the implementation of $\bar{f}$ that we presented in section 3.2 to downstream tasks and discard tokens both in fine-tuning and inference stages. The framework of PCAE for fine-tuning is illustrated in Figure 1 (b).

We observe two important differences between pre-training acceleration and downstream acceleration. First, downstream tasks are more sensitive to token reduction. The number of remaining tokens can be as less as 18 in the pre-training stage, but we observe serious performance degradation on downstream tasks in this extreme case. Therefore, we at most discard 40% tokens each stage and discard tokens after $0^{th}$, $7^{th}$, $10^{th}$ layer. Second, the discarded tokens could be used to improve performance. The discarded tokens are not used in the pre-training stage, but we find they are useful for downstream tasks. Therefore, we append the average of discarded tokens to the retained tokens in downstream tasks. However, we remove these appended tokens before final global pooling. Besides, following Liang et al. (2022), we apply a warmup strategy on the drop ratio parameter in fine-tuning. Specifically, we do not discard any token in the previous 10 epochs and then gradually increase the drop ratio to the target value in the following 25 epochs. Then, we keep the target drop ratio until fine-tuning ends.

Table 2: Object detection and instance segmentation on COCO with Mask R-CNN 1× schedule.

| Methods | Epochs | Object Detection | | | Instance Segmentation | | |
|---|---|---|---|---|---|---|---|
| | | $\text{AP}^b$ | $\text{AP}^b_{50}$ | $\text{AP}^b_{75}$ | $\text{AP}^m$ | $\text{AP}^m_{50}$ | $\text{AP}^m_{75}$ |
| *Supervised methods:* | | | | | | | |
| DeiT | 300 | 46.9 | 68.9 | 51.0 | 41.5 | 65.5 | 44.4 |
| *Contrastive learning methods:* | | | | | | | |
| MoCo v3 | 300 | 45.5 | 67.1 | 49.4 | 40.5 | 63.7 | 43.4 |
| DINO | 400 | **46.8** | 68.6 | 50.9 | **41.5** | 65.3 | 44.5 |
| *Masked Image Modeling methods:* | | | | | | | |
| BEiT | 300 | 39.5 | 60.6 | 43.0 | 35.9 | 57.7 | 38.5 |
| BEiT | 800 | 42.1 | 63.3 | 46.0 | 37.8 | 60.1 | 40.6 |
| MAE | 300 | 45.4 | 66.4 | 49.6 | 40.6 | 63.4 | 43.7 |
| MAE | 1600 | 48.4 | 69.4 | 53.1 | 42.6 | 66.1 | 45.9 |
| PCAE | 300 | 47.6 | 68.3 | 52.4 | 42.0 | 65.3 | 45.6 |
| PCAE | 800 | **48.8** | 69.6 | 53.6 | **43.1** | 66.9 | 46.4 |

## 4 EXPERIMENTS

This section is organized as follows: The comparisons between PCAE and related methods over multiple benchmarks including classification, and object detection are delivered in section 4.1. Section 4.2 exhibits the acceleration performance of PCAE in downstream tasks. Section 4.3 presents ablations on hyper-parameters of PCAE. The settings of data augmentation and optimization keep consistent with MAE (He et al., 2022). The mask ratio of 75% is applied on the input image across all experiments, and PCAE discards 50% tokens after $0^{th}$, $4^{th}$, and $8^{th}$ layer respectively in default. For more details please refer to the supplementary material.

### 4.1 COMPARISON WITH OTHER METHODS

**Fine-tuning on ImageNet-1k.** We follow the standard MIM evaluation protocol that fine-tunes the pre-trained model on the ILSVRC-2012 ImageNet for 100 epochs. We follow the fine-tuning schedule and hyper-parameters in MAE (He et al., 2022) out of fair comparison. Under this setting, we compare PCAE with its main counterpart MAE and other self-supervised methods.

Referring to Table 1, PCAE largely outperforms MAE with the same number of epochs but saves around 35% computational cost. For the same performance, PCAE only takes 1/8 overhead compared with MAE. Although BEiT (Bao et al., 2022) and CAE (Chen et al., 2022) can benefit from DALL-E (Ramesh et al., 2021) which is pre-trained on 250M images, PCAE still reach a better or comparable fine-tuning performance. Contrastive learning methods rely on complex data augmentation which badly increases the computational cost, especially for DINO (Caron et al., 2021) which relies on the multi-crop operation. As presented in Table 1, PCAE outperforms DINO in the fine-tuning benchmark at a much lower cost (around 1/12).

**Object detection and segmentation.** Besides the classification task, we evaluate PCAE on object detection and segmentation to demonstrate its potential in dense prediction tasks. In this setting, we fine-tune Mask R-CNN (He et al., 2017) with the pre-trained parameters in an end-to-end manner on COCO (Lin et al., 2014). The ViT backbone is adapted for use with FPN (Lin et al., 2017). We report the box AP for object detection and the mask AP for instance segmentation in Table 2. The results show that PCAE consistently outperforms other methods both in object detection and instance segmentation.

**Throughput comparison.** He et al. (2022) shows that MAE suffers serious performance degradation once its mask ratio exceeds 75%, which prevents MAE keep fewer tokens in the encoder. However, we surprisingly find that PCAE is robust to extremely high mask ratios and even works well with them. Specifically, as we presented in Table 4, the performance of PCAE is nearly not affected even when we mask 90% input patches. We owe this merit of PCAE to the relaxed reconstruction task. PCAE only reconstructs around 10% of patches, which makes the reconstruction task relatively simple, and thus the encoder could tolerate higher mask ratios. The relaxed reconstruction task benefits both the encoder and the decoder.

Table 3: Ablation study over drop stages (include 0) with a fixed drop ratio of 50%.

| Exp id | drop tokens | drop stage | drop ratio (%) | total (%) | GPU Day | Accuracy (%) |
|---|---|---|---|---|---|---|
| Exp id = 1 | × | 0 | 0 | 0 | 5.7 | **82.3** |
| Exp id = 2 | ✓ | 1 | 50 | 50 | 3.9 | 81.0 |
| Exp id = 3 | ✓ | 2 | 50 | 75 | 3.6 | 81.8 |
| Exp id = 4 | ✓ | 3 | 50 | 87.5 | 3.3 | **82.3** |
| Exp id = 5 | ✓ | 4 | 50 | 93.75 | 3.2 | 81.3 |

Table 4: The performance on ImageNet and COCO as well as the throughput of PCAE with different mask ratios.

| Mask ratio | 75% | 85% | 90% |
|---|---|---|---|
| FT Acc. | 83.6 | 83.6 | 83.5 |
| COCO Det. | 47.6 | 47.7 | 47.4 |
| Throughput | 550.5 | 739.7 | 876.1 |

Table 5: ImageNet classification under different drop ratios.

| Ratio | Throughput | MACs | Acc. |
|---|---|---|---|
| 0 | 286 | 17.57 | 83.8 |
| 0.1 | 330 (↑15%) | 15.34 (↓13%) | 83.7 (↓0.1) |
| 0.2 | 385 (↑35%) | 13.19 (↓25%) | 83.5 (↓0.3) |
| 0.3 | 449 (↑57%) | 11.29 (↓36%) | 83.2 (↓0.6) |
| 0.4 | 538 (↑88%) | 9.53 (↓46%) | 82.6 (↓1.2) |

Table 6: Ablation on drop stage with fixed total drop ratio of $87.5\%$.

| Exp id | drop stage | drop ratio | GPUd. | Acc. |
|---|---|---|---|---|
| Exp id = 1 | 1 | 87.5 | 3.0 | 81.5 |
| Exp id = 2 | 3 | 50 | 3.3 | **82.3** |
| Exp id = 3 | 6 | 29 | 3.5 | 82.0 |
| Exp id = 4 | 12 | 16 | 3.6 | 81.5 |

Table 7: Ablation on the totally drop ratio.

| Exp id | drop ratio | total | GPUd. | Acc. |
|---|---|---|---|---|
| Exp id = 1 | 37 | 75 | 3.6 | 81.8 |
| Exp id = 2 | 41.5 | 80 | 3.5 | **82.4** |
| Exp id = 3 | 47 | 85 | 3.4 | 81.9 |
| Exp id = 4 | 50 | 87.5 | 3.3 | 82.3 |
| Exp id = 5 | 53.6 | 90 | 3.3 | 81.7 |

This feature enables PCAE further to accelerate the MIM task. We report the throughput of PCAE on V-100 32G in Table 4 with different mask ratios. The results show that PCAE handles at most 739.7 images per second without sacrificing performance, which is 2.25 times the throughput of MAE (739.7 img/s vs 328.4 img/s). We further adapt PCAE to SimMIM, termed PCAE-SimMIM, which accelerates the throughput 2.37 times with comparable performance. For more details please refer to the supplementary.

## 4.2 DOWNSTREAM ACCELERATION

Model acceleration is critical for the practical deployment of vision transformers. Due to the simplicity and flexibility of the proposed progressive token reduction strategy, it can be easily incorporated into the supervised setting and benefit the inference speed. With the pre-trained model, we fine-tune it on ImageNet under different drop ratios for 100 epochs and report the Top-1 accuracy. As presented in Table 5, PCAE balance the acceleration and performance well. Specifically, PCAE accelerates 57% throughput while the performance drop is within 0.6%. Even for the extreme case where only 42 tokens are kept out of 196 tokens, the performance slightly drops 1.2% while the throughput improves 88%. These results further demonstrate the effectiveness of PCAE as it manages to accelerate different scenarios including self-supervised pre-training, supervised fine-tuning, and inference.

## 4.3 ABLATION STUDY

We apply the same setting for all ablation studies unless specified. Specifically, we pre-train PCAE on ImageNet for 100 epochs with batch size of 1024 and fine-tune the pre-trained model on ImageNet for 100 epochs with the default schedule of MAE for a fair comparison.

Initially, we highlight critical hyper-parameters of PCAE for clarity. Referring to Eq. 2, our discarding strategy is determined by $I$ which decides where tokens are discarded and the drop ratio. As the value scope of $I$ is too huge, we decompose it into two parameters, i.e., drop stage and drop

Table 8: We fix the drop stage and drop ratio, and ablate on different drop cases,include the comparison to randomly drop, *abbr*. RD. The figure layout in right shows the drop position.

| Exp id | RD | drop case | GPUd | Acc. |
|---|---|---|---|---|
| Exp id = 1 | ✓ | 1 | 3.3 | 81.0 |
| Exp id = 2 | × | 1 | 3.3 | **82.3** |
| Exp id = 3 | × | 2 | 3.7 | 82.0 |
| Exp id = 4 | × | 3 | 3.4 | 82.0 |
| Exp id = 5 | × | 4 | 3.1 | 81.5 |
| Exp id = 6 | × | 5 | 4.8 | 80.7 |

case. The drop stage indicates how many times we should discard tokens, i.e., $card(I)$. Drop case indicates where we prefer to discard tokens, i.e., shallower layers, deeper layers, etc. (refer to Table 8 for more details). In summary, drop stage, drop case, and drop ratio are the three critical hyperparameters for PCAE. In the following, we ablate on different settings of these hyper-parameters to show their effects.

**How many tokens should we discard?** We first fix the drop ratio and drop case, and ablate on the drop stage. As presented in Table 3, the results show that removing redundant tokens from the prediction target significantly improves efficiency (exp 1 and others). Among these experiments, we find that the reconstruction targets prefer to moderate drop ratio, which is similar to the input end, and discarding 87.5% tokens (exp 4) achieves the best balance between performance and efficiency.

We further control the total drop ratio equal to 87.5% and ablate on the drop stage, i.e., we need to drop tokens in few times with a large drop ratio or more times with a smaller drop ratio. A large drop ratio improves efficiency but may suffer performance degradation, and vice versa. The empirical results in Table 6 show that dropping 3 times is the best choice to balance them.

Then, we fix the drop stage as 3 and grid search for the best total drop ratio from 75% to 90%. As presented in Table 7, totally discarding 80% tokens reaches the best performance. However, compared with experiment 4, we find that discarding 50% tokens three times is more simple and more efficient and only slightly degrades the performance. Thus, we apply the configuration of exp 4 in default.

**Where should we discard tokens?** We ablate on the drop case parameter here. We propose five different cases to discard tokens and their details are illustrated at the right of Table 8. The results in Table 8 show that evenly allocating the discarding operation among blocks performs better than continuously discarding tokens (compare exp 2, 3, 4 with exp 5, 6). The drop case 1 performs best compared with case 2 and case 3 (exp 2, 3, 4), which indicates that token reduction supposes to be avoided at the last layer.

We also compare with the randomly drop in Table 8, and the result shows that the performance of randomly discarding tokens far falls behind that of our strategy (exp 1 and exp 2). Besides, we provide more performance of randomly drop under different configurations in the supplementary, and results show the proposed strategy outperforms it in all cases.

## 5  CONCLUSION

In this paper, we present PCAE which addresses the problem of the redundant reconstruction target in MIM methods. Experimental results show that PCAE outperforms MAE over multiple benchmarks even only reconstructs around $10\%$ tokens. The lightweight reconstruction task enables PCAE to achieve comparable fine-tuning performance to MAE with only 1/8 GPU days. We visualize the progressive token reduction process and find that PCAE manages to retain semantic-abundant components in images. PCAE exhibits elegant symmetry at the input and output end as both of them are compressed. What beyond the proposed method is the insight that MIM-based methods benefit from more compact prediction targets. We wish it can inspire future works.

**Acknowledgement** This work was supported in part by the National Natural Science Foundation of China under Grant 62250055, Grant 61932022, Grant 61931023, Grant 61971285, Grant 61831018, Grant 61871267, Grant 61720106001, Grant 62120106007, Grant 61972256, Grant T2122024, Grant 62125109, and in part by the Program of Shanghai Science and Technology Innovation Project under Grant 20511100100.

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

# APPENDIX

## A DIVERSITY MEASUREMENT

Initially, we show how we measure diversity. We start by looking at the input token sequence from a patchify image. Denote the number of tokens as $N$, and the dimension as $D$, the token sequence can be written as a matrix $A \in \mathbb{R}^{N \times D}$. Following the common practice in self-supervised learning Wang & Isola (2020); Ermolov et al. (2021), we analyze the token representations on the unit sphere, i.e., we normalize the length of each token to one. As the convex envelope of matrix rank, the nuclear norm is widely used to measure the diversity of matrix rows Cui et al. (2020). Specifically, according to the theorem in Fazel (2002), $\|A\|_*$ is the convex envelope to $rank(A)$ when $\|A\|_F \leq 1$. Since $\|A\|_F = \sqrt{N}$ (note that each row vector in $A$ is unit vector), the convex envelope of $rank(A)$ turn to $\|A\|_*/\sqrt{N}$. According to the property of the nuclear norm, $\|A\|_*/\sqrt{N}$ is bounded by $\sqrt{k}$ where $k = min(N, D)$. In practical implementation, $N$ is the length of the token sequence and usually is 196, and $D$ is the embedding dimension and usually is 768 or 512. Thus, $min(N, D) = N$ and $\|A\|_*/\sqrt{N} \leq \sqrt{N}$. Since the length of token sequences varies during the forward process and the upper bound of $\|A\|_*/\sqrt{N}$ depends on $N$, we instead apply $\|A\|_*/N$ which has a clear upper bound and get rid of the influence from $N$ to measure the diversity in token sequences.

## B IMPLEMENTATION DETAILS

**Architecture.** We study the ViT base architecture (12 transformer blocks with dimension 768) and follow the implementation of Dosovitskiy et al. (2021). The teacher and the student branch share the same architecture. We follow the implementation of He et al. (2022) for the decoder part. Specifically, we apply an FC (fully-connected layer) as the dimension adapter between the encoder and decoder and two transformer blocks to decode the masked tokens with the output of the student. At last, we apply another FC to convert the decoder dimension into the encoder dimension for reconstruction. What is different from MAE is that we only reconstruct tokens output by the teacher.

**Hyper-parameters.** We follow the setting of hyper-parameters of MAE (He et al., 2022). For the EMA operation, we apply the fixed momentum equal to 0.9999.

## C THE NAIVE APPROACH OF DISCARDING PART OF THE TARGETS

Table 9: Performance comparison.

| Setting | Acc |
|---|---|
| Base | **82.1** |
| Setting id = 1 | 79.6 |
| Setting id = 2 | 79.9 |

Table 10: The comparison between SimMIM and PCAE-SimMIM.

| Method | FT acc | GPUd | Throughput |
|---|---|---|---|
| SimMIM | 82.1 | 6.8 | 204.9 |
| PCAE-SimMIM | 82.3 | 3.7 (↓45.6%) | 485.4 (↑137%) |

We present the naive approach of discarding part of the reconstruction targets of MAE here. We discard patches with the proposed strategy and only reconstruct remnant patches with MAE. We apply two settings for the base MAE. In the first setting, we simply mask gradients from discarded patches while we only decode remnant tokens in the decoder in the second setting. We observe that both settings suffer serious performance degradation as presented in Table 9. This demonstrates what we stated in the main text that simply discarding part of the reconstruction target damages useful information as well as eliminates redundancy.

## D ADAPT TO SIMMIM

For SimMIM, which forwards visible tokens together with the mask tokens at the encoder, we can regard the encoder in SimMIM as the decoder in MAE, and thereby speed up it via discarding most

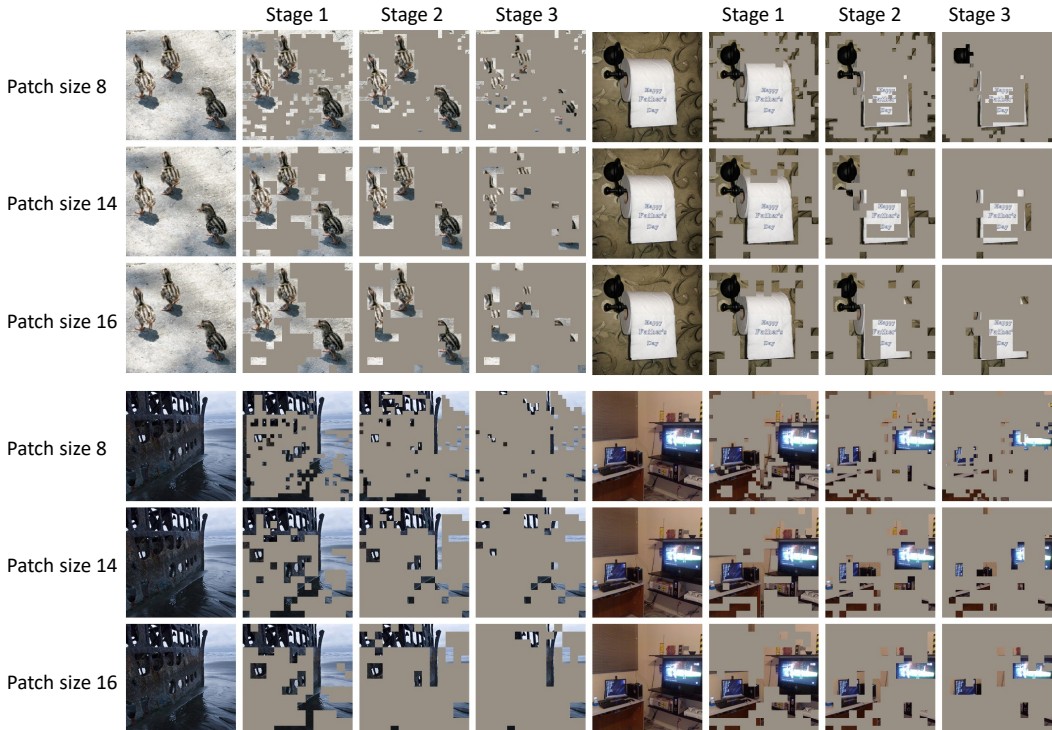

Figure 4: The visualization of PCAEunder different patch sizes. Images are arranged from left to right, top to bottom. We forward images with PCAE pre-trained model and record the position of the remaining tokens. Then, we mask patches corresponding to the discarded tokens at each stage. Note that we use models pre-trained on ImageNet100, so the visualization to patch size 16 is slightly different from that in the main text.

Table 11: The throughput comparison between PCAE-SimMIM and SimMIM. We adapt PCAE to SimMIM and compare its throughput with the original SimMIM on the same machine equipped with V100 GPU under different batch size. The acceleration is calculated under the same batch size if possible and compared with the maximum throughput otherwise.

| Batch size | PCAE-SimMIM | | | SimMIM | | |
|---|---|---|---|---|---|---|
| | Memory | Throughput | Accelerate | Memory | Throughput | Accelerate |
| 64 | 5.63G | 310.2 | 1.71× | 11.32G | 181.1 | 1.0× |
| 128 | 8.43G | 383.4 | 1.87× | 20.82G | 204.9 | 1.0× |
| 164 | 9.99G | 417.7 | 2.91× | 26.16 | 143.5 | 1.0× |
| 256 | 14.08G | 432.8 | 2.11× | / | / | / |
| 512 | 25.37G | 485.4 | 2.37× | / | / | / |

of the mask tokens. Specifically, the length of the token sequence is 196 (78 visible tokens and 118 mask tokens) in vanilla SimMIM, and we can speed up it by building a much more compact target for it. Then, the length of the token sequence can be reduced to 96 (78 visible and 18 mask tokens), thereby improving the efficiency. We adapt PCAE to SimMIM with minimum modification. Specifically, We run SimMIM for 100 epochs with the officially released code on a single machine with 8 V-100 and record the practical GPU day. We adapt PCAE to SimMIM with minimal modification and run PCAE-SimMIM for 100 epochs following the configuration of SimMIM.

Besides, we also report the throughput and memory usage in Table 11 by running on a single V-100. As presented in Table 10, we find that PCAE-SimMIM achieves comparable performance (PCAE-SimMIM 82.3 vs SimMIM 82.1 100 epoch) to SimMIM with 55% training hours (PCAE-SimMIM 11.1h vs SimMIM 20.3h on 8 V-100 machine). Besides, we provide the throughput comparison

Table 12: The comparison between PCAE and other strategies.

| Setting | PCAE | Token-Importance | PCAE-Pretrained | Saliency-Pretrained |
|---|---|---|---|---|
| FT accuracy | **82.3** | 80.7 | 83.1 | 82.7 |

Table 13: The comparison between the proposed redundant token identification strategy and randomly drop (RD)

| Method | drop stage | drop ratio | drop case | FT acc |
|---|---|---|---|---|
| PCAE | 3 | 0.5 | 1 | **82.3** |
| RD | 3 | 0.5 | 1 | 81.0 |
| RD | 3 | 0.5 | 2 | 81.1 |
| RD | 3 | 0.5 | 3 | 80.3 |
| RD | 6 | 0.29 | 1 | 80.4 |
| RD | 12 | 0.16 | 1 | 81.0 |

Table 14: Performance over different architectures.

| Method | architecture | epoch | FT acc. |
|---|---|---|---|
| MAE | ViT-S | 300 | 80.8 |
| MoCo v3 | ViT-S | 300 | 81.7 |
| BEiT | ViT-S | 300 | 81.7 |
| PCAE | ViT-S | 300 | **81.9** |
| MoCo v3 | ViT-L | 300 | 84.1 |
| MAE | ViT-L | 400 | 84.3 |
| PCAE | ViT-L | 300 | **85.1** |

between PCAE-SimMIM and SimMIM below, which shows PCAE still works well when transferred to other methods.

# E COMPARISON WITH OTHER STRATEGIES

We consider two settings here. In the first setting, we measure the token importance with the average attention from other tokens. Specifically, we average the attention matrix of different heads in the self-attention block and calculate the mean of each column to get the importance of each token. Then we discard less important tokens. We follow the same implementation and settings with PCAE. We term this setting as Token Importance. In the second setting, we replace the momentum encoder of PCAE with a DINO pre-trained ViT-S model, and identify redundant tokens according to the saliency produced by the pre-trained model or PCAE respectively for comparison. We term this setting as Saliency-Pretrained or PCAE-Pretrained. We run these settings for 100 epochs and report their fine-tuning accuracy in Table 12.

The results show that Token-Importance performs much worse compared with PCAE (80.7 vs 82.3). We owe it to the lack of supervisory signals. The network is hard to identify important tokens at the beginning stage where parameters are randomly initialized, and the worse beginning propagates errors to later phases, resulting in serious performance degradation ultimately. We also find that PCAE-Pretrained performs better than Saliency-Pretrained. It is caused by the difficulty in identifying redundant tokens at intermediate layers, especially at shallow layers for the saliency-based method, and thus it may discard important tokens and causes performance degradation.

# F COMPARISONS WITH RANDOMLY DROP

We supplement more experiments here to show our advantage over randomly drop (RD). For a fair comparison, we fix the total drop ratio equal to 87.5% and search for the best strategy for randomly discarding

Table 15: Ablation study on decoder depth.

| Decoder depth | 2 | 4 | 6 | 8 |
|---|---|---|---|---|
| FT Acc. | 83.6 | 83.6 | 83.6 | 83.6 |
| COCO Det. | 46.9 | 46.9 | 47.6 | 47.6 |

by the drop case and drop stage. Specifically, we fix the drop stage and run the randomly drop with different drop cases (we select drop cases with better results by referring to Table 8). The results in Table 13 show that the randomly drop is much worse than PCAE. Besides, we also fix the drop case and search on the drop stage, but the results still show that PCAE outperform the randomly drop by a large margin.

# G VISUALIZATIONS UNDER DIFFERENT PATCH SIZES

We show the visualization to PCAE under different patch sizes. The patch size determines the information amount of each token, and thus has influences on the discarding strategy in PCAE. We

pre-train models on ImageNet100 for 100 epochs under different patch sizes, and show their visualization in Figure 4. The visualization indicates that PCAE discards similar content with different patch sizes.

## H    MORE ABLATION STUDY

In this section, we present more ablation studies of PCAE.

**Decoder depth.** We ablate the decoder depth of PCAE. The results in Table 15 show that the fine-tuning performance of PCAE is robust to decoder depth but the performance in downstream tasks relies on deeper decoders, which is consistent with MAE (He et al., 2022).

**Architectures.** We run PCAE over different architectures with 85% mask ratio. We apply the same configuration on hyper-parameters across different architectures and the results in Table 14 show that PCAE works well over different architectures, which demonstrates that the hyper-parameters are robust to model size.

