# OpenReview forum: "Progressively Compressed Auto-Encoder for Self-supervised Representation Learning"
_ICLR.cc/2023/Conference — ICLR 2023 poster_

### Official Review · Reviewer_qAL9 · 2022-10-23

**Confidence:** 3
**Correctness:** 4
**Technical Novelty And Significance:** 3
**Empirical Novelty And Significance:** 3
**Recommendation:** 6

**Clarity, Quality, Novelty And Reproducibility:**

The writing part of this paper is good and easy to follow. However, the formulation of the framework is not rigorously presented, which has been pointed out in the previous box.

**Strength And Weaknesses:**

Strength:

1.  This paper addresses the patch redundancy problem in MIN, which was largely overlooked in previous literature.

2. The paper is well-written and easy to follow.

Weaknesses:

1. It seems unclear to me where speed-up comes from. It seems to me that speed-up mainly comes from reducing the decoding overhead by decreasing the number of tokens to be reconstructed. However, the decoder of the original MAE has been kind of lightweight. Even more, following papers like SimMIM, CAE, etc, have shown that a one-layer decoder can achieve nearly the same performance.  In this regard, do the authors still believe their proposed methods are practical and significant compared to just continue reducing the capacity of the decoder, e.g., a one-layer decoder?

2. There are some issues with the formulations.
- Equation (1) is not rigorously presented: (i) The dimension of w and p is not aligned. One is “a” shared token while the other one is “a sequence of” tokens; (ii) The authors claim that the M function is a similarity function. But one of its inputs are the embeddings of masked tokens and the other input is a scalar k. What does their similarity mean? (3) So, what is the objective this equation wants to maximize?

- Around Equation (2), please clarify the NxD. Does N mean the number of remaining tokens at every layer? If so, then a fixed N cannot be used since the authors change it at some selected layers.

- Equation (3) has a minor error. It should be summing over the j.


**Summary Of The Paper:**

This paper aims to reduce the pre-training overhead of MAE by decreasing the number of tokens to be reconstructed. Specifically, the authors follow CAE to design a momentum encoder to operate on masked tokens and gradually discard some tokens (that are closer to the average tokens) at some layers. In this way, the overhead of decoding is reduced, since fewer tokens are needed to be predicted. This paper claims that this manner can speed up the pre-training without compromising performance on different downstream tasks.

**Summary Of The Review:**

This paper proposes to speed up the decoding of MAE by reducing the number of tokens to be predicted, which is interesting. However, It seems that the current technique can only be applied in MAE since it mainly reduces the decoding part. For other methods such as BEiT, CAE and SimMIM have more lightweight decoding, it seems not that suitable.

---

> ### Author Response · Authors · 2022-11-09
> **Response to Reviewer qAL9**
>
> We appreciate your insightful comments and valuable suggestions, and we respond to your concerns in the following:
>
> - **Q1: The speed-up of PCAE**
>
>     As we posted in the general response, PCAE accelerates MIM methods by relaxing the reconstruction task whether the method uses or not the decoder. **For methods like MAE**, PCAE accelerates them by reducing the number of masked tokens in the decoder. Although the decoder has fewer parameters compared with the encoder, it operates 100% of input tokens while the encoder only operates 25% of tokens, and thus the overhead of the decoder could exceed that of the encoder (5.26G vs 4.3G for MAE ViT base). We find that the reconstruction for around 10% of tokens is sufficient to achieve comparable performance to MAE.
>     **For methods like SimMIM**, PCAE accelerates them by reducing the number of masked tokens in the encoder, and SimMIM even benefits more from PCAE as the encoder is more complicated (for more details please refer Appendix D in the revision). Both paradigms significantly improve the efficiency of MIM methods.
>
>     Besides, we also find that PCAE is robust to extremely high mask ratios. This merit enables PCAE further accelerates MIM methods by applying higher mask ratios on the input images.
>
> - **Q2: issues in formulations**
>
>     (1) We are sorry for this mistake and will fix the description to $w$ in our revisions.
>
>     (2) We find we wrote an extra parenthesis in equation 1, which leads to an ambiguity. The correct one suppose to be $\max\limits_{\theta, \phi, \omega} \mathop{\mathbb{E}}\limits_{x\sim \mathcal{X}} \mathcal{M}(\bar{f}((1 - M)\odot x; k), g_\phi( h(f_\theta(M\odot x), \omega + p))$. $\mathcal{M}$ measure the similarity of the output of $\bar{f}$ and $g_{\phi}$, i.e., $\mathcal{M}(\bar{f}(\cdot;\cdot), g_\phi(\cdot, \cdot))$. We are sorry for this typo and will fix it in our revisions.
>
>     (3) This objective aims to maximize the similarity between the output of the teacher and the output of the decoder, and it serves as the loss function for the reconstruction task.
>
>     (4) Yes, $N$ indicates the number of remaining tokens. Thanks for your patience, and we will add subscript to it to indicate it is various in different layers.
>
>     (5) Thanks! We will fix it in our revisions.
>
> We sincerely thank for your efforts in reviewing this paper, and we are looking forward to your response.

---

> ### Author Response · Authors · 2022-11-18
> **Thanks and Looking Forward to Your Reply**
>
> Dear Reviewer,
>
> We would appreciate your valuable suggestions which helps us a lot in improving our paper. We wonder if our response has cleared your concerns and would like to answer additional questions at any time.
>
> Many thanks for your comments again.

---

### Official Review · Reviewer_Z7NJ · 2022-10-23

**Confidence:** 4
**Correctness:** 3
**Technical Novelty And Significance:** 3
**Empirical Novelty And Significance:** 2
**Recommendation:** 6

**Clarity, Quality, Novelty And Reproducibility:**

+ **Clarity**: The paper is easy to follow despite that some claims of the motivation need clarification (see Weaknesses).
+ **Quality**: The paper addresses a practical problem in an intuitive way, but the experimental results seem insufficient to fully validate the method.
+ **Novelty**: The method is new to the community of MIM. But it should be better to discuss some works on manipulating tokens in ViT architectures, e.g., "Not All Tokens Are Equal: Human-centric Visual Analysis via Token Clustering Transformer" (CVPR'22).
+ **Reproducibility**: I did not find the details of implementation, e.g., hyper-parameters.

**Strength And Weaknesses:**

### Strengths
1. **Motivation**: It is an important task to reduce the computation overhead in MIM pre-training. And the introduced method seems practical in the real world.
2. **Writing**: The paper is generally well-written and easy to follow.
3. **Method design**: The introduced method for redundant token identification is easy to implement and makes sense.
4. **Results**: Competitive results on ViT-Base are shown.

### Weaknesses
1. **Motivation**: Though I get the overall point of speeding up MIM pretraining, I am confused by some of the declarations. (1) In the first paragraph of the introduction, the authors pointed out that the efficiency bottleneck lies on the decoder, which is limited to the encoder-decoder architectures like MAE while neglecting the other ones like BEiT and SimMIM (unify encoder and decoder). Moreover, the number of layers in the decoder should influence the accuracy and efficiency of PCAE, which should be analyzed. (2) In the second paragraph of the introduction, the authors claimed that useful information is discarded when naively removing part of the reconstruction patches. The claim is ungrounded and the authors should provide some evidence for it. Intuitively, useful information should be learned in the unmasked tokens and have nothing to do with removing part of the masked tokens.
2. **Method design**: Though the introduced method for redundant token identification (removing the ones close to mean tokens) is intuitive and indeed helps training. The authors are recommended to analyze other alternative options for it.
3. **Results**: (1) The experiments on more architectures, e.g., ViT-S/L, are necessary for verifying the generalization ability of the introduced method. Especially since the method depends on the drop ratio and drop layers, which might "overfit" on the architecture and make it hard to deploy on other models. (2) As the method can save time, it would be interesting to see if better results can be achieved by training with more data or more epochs while the overall time is still less than other MIM methods. (3) In the appendix, the authors introduce to adapt PCAE on top of SimMIM but did not provide accuracies. Will PCAE perform well on SimMIM?

**Summary Of The Paper:**

The paper aims to design a more efficient MIM method by taking visual redundancy into account. The paper identifies the redundant tokens and progressively reduces the number of tokens for the reconstruction target. Competitive results are achieved while accelerating MAE for 1.9 times.

**Summary Of The Review:**

Overall, I am intrigued by the motivation for this work and believe it is important for practical applications. However, there are still some issues, see Weaknesses.

---

> ### Author Response · Authors · 2022-11-12
> **Response to Reviewer Z7NJ part I**
>
>
> We appreciate your insightful comments and valuable suggestions, and we respond to your concerns in the following:
>
> - **Q1: The motivation of PCAE**
>
>     (1) **The efficiency bottleneck of MIM methods**
>
>     We want to clarify that the efficiency bottleneck of MIM methods lies on the reconstruction task. Referring to the context, we take MAE as an example to explain the reconstruction task in MIM, and the statement that "the efficiency bottleneck lies on the decoder" only refer to MAE.
>     As we posted in the general response, PCAE can accelerate SimMIM alike methods by reducing the number of masked tokens in their encoders and performs well in unify encoder-decoder architectures. For methods using extra pre-trained models like BEiT, the momentum encoder can be replaced with the pre-trained model, and PCAE can be easily adapted to them with little modifications.
>
>     (2) **The ablation study on the decoder depth**
>
>     We report the impact of decoder depth to FT acc and the mAP on COCO detection in the following table. All results are obtained from 300 epoch pre-training.
>
>     |Decoder depth | FT acc    | COCO det   |
>     |:------------:|:---------:| :---------:|
>     |    2         | 83.6      | 46.9       |
>     |    4         | 83.6      | 46.9       |
>     |    6         | 83.6      | 47.6       |
>     |    8         | 83.6      | 47.6       |
>
>     As presented in the table above, PCAE is robust to the decoder depth in terms of fine-tuning accuracy, but deep decoder will benefit its performance in detection task.
>     Since PCAE works well even better with extremely high mask ratios (for more details please refer to the general response), we further report the acceleration under higher mask ratios. We take the maximum throughput of MAE on single V100 as baseline and report the acceleration.
>
>     |Decoder depth  | mask 85%   | mask 90%   |
>     |:------:       | :---------:|:----------:|
>     |2              | 2.87       |3.24        |
>     |4              | 2.66       |3.02        |
>     |6              | 2.46       |2.86        |
>     |8              | 2.25       |2.67        |
>
>     (3) **Justification for "useful information is discarded when naively removing part of the reconstruction patches"**
>
>     We think there may be some misunderstandings here. This sentence we mean that part of the supervisory signals (for example, patches that need to be recovered in MAE) are discarded when we naively remove part of the reconstruction targets, and thus the model learn less information from the reconstruction task and produce deteriorated representations. Note that we refer to the reconstruction targets which supervise on the self-supervised pretext rather than masked tokens which does not provide any extra information. We ground it through the ablation study in Table 9 where we remove part of the reconstruction targets using the proposed strategy, and the results show that it suffers serious performance degradation (82.1 -> 79.9). We sorry for this ambiguous sentence and will make it more clear in later revisions.
>
> - **Q2: Other alternative options for removing redundant tokens**
>
>     Thanks for your valuable suggestion! We try two options to remove redundant tokens and post their performance and analysis to them in the **General response Q3**. Please refer to it for more details. In summary, the results show that PCAE performs better than these options given the unsupervised setting.

---

> > ### Author Response · Authors · 2022-11-12
> > **Response to Reviewer Z7NJ part II**
> >
> >
> > - **Q3: More results**
> >
> >     **(1) The performance on different architectures.**
> >
> >     We apply the same drop ratio, drop case and drop stage as ViT-B for ViT-S/L and report their performance under 85% mask ratio (as we posted in the general response, we find PCAE works well even better with 85% mask ratio, which further accelerate training) in the following.
> >
> >     |Methods    | architecture    | epoch      | FT acc     |
> >     |:------    |:---------------:| :---------:|:----------:|
> >     |MAE       | ViT-S           | 300        | 80.8       |
> >     |MoCo v3    | ViT-S           | 300        | 81.7       |
> >     |BEiT       | ViT-S           | 300        | 81.7       |
> >     |PCAE       | ViT-S           | 300        | **81.9**   |
> >
> >     |Methods    | architecture    | epoch      | FT acc     |
> >     |:------    |:---------------:| :---------:|:----------:|
> >     |MoCo v3    | ViT-L           | 300        | 84.1       |
> >     |MAE        | ViT-L           | 400        | 84.3       |
> >     |PCAE       | ViT-L           | 300        | **85.1**   |
> >
> >     The results show that PCAE still be competitive in other architectures, and the hyper-parameters generalize well across architectures.
> >
> >     **(2) Training with more epochs or data.**
> >
> >     As we posted in the general response, PCAE not only speeds up training but also converges much faster than MAE. We find that PCAE achieves comparable performance to MAE 1600 epoch with only 300 epochs. We further run PCAE for more epochs and find that PCAE can outperforms MAE when trained with 800 epochs. The overall overheads of PCAE is far less than that of MAE. Thus, PCAE is able to outperforms other MIM methods with much less overheads.
> >
> >     A recent study [1] shows that pre-training with high-resolution input can benefit dense prediction tasks. Therefore,  we consider training PCAE with high-resolution images. We adjust the input size of PCAE to 320 for a similar throughput to MAE. Then, we run PCAE for 300 epochs and evaluate its performance on ImageNet fine-tuning, COCO detection and instance segmentation. The results in the table below show that PCAE could achieve better performance on dense prediction tasks with similar overheads by pre-training on high-resolution inputs.
> >
> >     |Method | input size        | throughput        | FT acc.     | COCO det. | COCO seg. |
> >     |:------    | :---------:|:----------:|:--------:|:--------:|:--------:|
> >     |MAE      | 224       | 328.4       | 82.9     | 45.4      | 40.6 |
> >     |PCAE       | 224       | 641.1       | 83.6     |  47.7  | 42.2 |
> >     |PCAE       | 320       | 331.3       | 83.6     |  48.1  | 42.5 |
> >
> >     **(3) The performance of PCAE-SimMIM**
> >
> >     We report the performance of PCAE-SimMIM in Appendix D. Specifically, we adapt PCAE to SimMIM with minimal modification and find that PCAE-SimMIM achieves comparable performance to SimMIM (PCAE-SimMIM 82.3 vs SimMIM 82.1 100 epoch).
> >
> > - **Q5: Reproducibility**
> >
> >     We provide our implementation details in the appendix section B and feel sorry for you missing it. Specifically, we follow the implementation of MAE and apply its optimization hyper-parameters including learning rate, schedule, and optimizer for fair comparisons. We set fix momentum equal to 0.9999. We will also release our code and pre-trained models once this paper is accepted by ICLR2023 to ensure its reproducibility.
> >
> > - **Q6: Comparison to related work**
> >
> >     Thanks for your suggestion! [2] considers the patch importance in human-centric visual tasks and propose to merge patches shared similar semantic concepts while keep fine resolution for regions containing critical details. [2] works with architectures specially designed for human-centric tasks and require supervisory signals, and thus is inaccessible to self-supervised learning where the architecture is assumed to be task agnostic and supervisory signals are unavailable.  Therefore, we stage [2] as an important branch for dynamic vision transformer in dense prediction task. In contast, PCAE focus on self-supervised learning and does not rely on supervisory signals or specially designed architectures.
> >
> >     We will add [2] to our related work and discuss it in our revisions.
> >
> > We sincerely thank for your efforts in reviewing this paper, and we are looking forward to your response.
> >
> > [1] Xu, Yifan, et al. "Evo-vit: Slow-fast token evolution for dynamic vision transformer." Proceedings of the AAAI Conference on Artificial Intelligence. Vol. 36. No. 3. 2022.
> >
> > [2] Zeng, Wang, et al. "Not All Tokens Are Equal: Human-centric Visual Analysis via Token Clustering Transformer." Proceedings of the IEEE/CVF Conference on Computer Vision and Pattern Recognition. 2022.
> >
> > [3] Hu, Ronghang, et al. "Exploring Long-Sequence Masked Autoencoders." arXiv preprint arXiv:2210.07224 (2022).

---

> ### Author Response · Authors · 2022-11-19
> **Thanks and Looking Forward to Your Reply**
>
> Dear Reviewer,
>
> We would appreciate your valuable suggestions which helps us a lot in improving our paper. We wonder if our response has cleared your concerns and would like to answer additional questions at any time.
>
> Many thanks for your comments again.

---

> > ### Comment · Reviewer_Z7NJ · 2022-11-24
> > **Thanks for the authors' efforts**
> >
> > I confirm I have read the authors' response and appreciate the authors' efforts on additional experiments, especially the interesting high-resolution ones. Most of my concerns have been addressed, but I agree with other reviewers on the marginal improvements of PCAE, given the newly added comparison with SimMIM and experimental results with longer epochs and other architectures. Despite the weakness above at this moment, I consider PCAE's contribution to reducing the computational overhead in MIM pre-training is practical and helpful. So I would like to weakly accept this work and maintain my original justification (6).

---

### Official Review · Reviewer_MSzM · 2022-11-02

**Confidence:** 3
**Correctness:** 4
**Technical Novelty And Significance:** 2
**Empirical Novelty And Significance:** 2
**Recommendation:** 6

**Clarity, Quality, Novelty And Reproducibility:**

Exploiting the redundancy in tokens to accelerate training is not novel, but the paper apply such idea to the self-supervised regime. A large amounts of experiments are conducted to support the claim. The paper is generally easy to follow.

**Strength And Weaknesses:**

Strength:
- The overall idea of improving the efficiency of MAE training by reducing the redundancy in reconstruction is reasonable.
- Large amount of experiments and ablations(drop ratio, drop location etc) to validate the claim.

Weaknesses:
- The proposed emperical criterior of 'similarity to mean token' is not compared with any other strategy. The author mention the potential to use matrix decomposition but no experimental results are shown.
- How are the token discarded in pretraining phase is not very clear. Is there also a warmup strategy for token discarding in pre-training?
- There is no comparison to other approach that dynamically discard token. Since the presented method can also be applied to supervised task, there should be room for comparison?

Other comments:
- The table number are incorrect. For example, you said "As the training may be influenced by many factors, we show the throughput comparison between PCAE and MAE in Table 4.2". But I guess you really mean Table 3.
- In the second paragraph of section 3.2, consider use a different symbol for the redundant token identification function? As D is already used in the definition of x_i.

**Summary Of The Paper:**

This paper improves the efficiency of MAE training by exploiting the redundancy in reconstructed tokens. They propose to use the similarity to mean token as an emprical criterior to prune the tokens, which leads to savings in computation. They achieve 1.5x-1.9x speed up compared with MAE, while maintain similar performance.

**Summary Of The Review:**

The method this paper presented is empirical but a large amounts of experiments are conducted to validate the effectiveness. The results of introducing 1.5x-1.9x speed up over MAE is somewhat significant afaik.

Post rebuttal:
I appreciate the response and additional exp results from the author. The exp results are solid but the relatively weak novelty keeps me from giving a higher rating.

---

> ### Author Response · Authors · 2022-11-10
> **Response to Reviewer MSzM**
>
> We appreciate your insightful comments and valuable suggestions, and we respond to your concerns in the following:
>
> - **Q1: Compare with other strategy**
>
>     As we mentioned in the third paragraph, the matrix decomposition exhibits prohibitive complexity in practice (slowing down the training 5 times), and thus we do not take it into consideration. It is hard to efficiently identify redundant tokens in cases where the supervisory is absent, and previous methods are only accessible for the supervised setting (please refer to the third paragraph in the related work section). Here we further consider two options to discard redundant tokens and post them in the **General response Q3**. The results show that PCAE performs better than other options under the unsupervised setting. We will add this discussion to our revisions.
>
> - **Q2: The token reduction in the pretraining phase**
>
>     Referring to Figure 1, we perform the token reduction in the teacher branch. Specifically, we discard 50% of tokens according to the proposed scoring metric at the 0th, 4th, and 8th layer three times. In this way, we reduce the token number from 147 to 18 ultimately. We find this strategy is robust and does not need any warmup strategy.
>
> - **Q3: Compare to supervised methods**
>
>     We want to clarify that the proposed PCAE is designed for self-supervised learning as it works without any supervisory signals. Here we show the comparison between PCAE and the SOTA dynamic ViT methods E-ViT [1]. Note that this comparison is unfair for PCAE as it does not use supervisory signals, and we report it here only for reference. We use the officially released code of E-ViT as codebase and replace its strategy with PCAE. All other settings follow the configuration of E-ViT.
>
>     |keep ratio | 0.9        | 0.8        |
>     |:------    | :---------:|:----------:|
>     |E-ViT      | 79.9       | 79.7       |
>     |PCAE       | 79.9       | 79.8       |
>
>     Note that the keep ratio in the above table indicates the ratio we keep tokens each time, and we follow the setting of E-ViT discard tokens three times. The real keep ratio in the last layer is 0.9^3 or 0.8^3.
>     The table above shows that PCAE is comparable to the SOTA supervised method, which demonstrates the robustness and generalization of PCAE.
>
> - **Q4: Other comments**
>
>     We are sorry for these typos and will fix them in our revisions.
>
> - **Q5: Reproducibility**
>
>     We provide our implementation details in the appendix section B and feel sorry for you missing it. Specifically, we follow the implementation of MAE and apply its optimization hyper-parameters including learning rate, schedule, and optimizer for fair comparisons. We set fix momentum equal to 0.9999. We will also release our code and pre-trained models once this paper is accepted by ICLR2023 to ensure its reproducibility.
>
> We sincerely thank for your efforts in reviewing this paper, and we are looking forward to your response.
>
> [1] Youwei Liang, Chongjian Ge, Zhan Tong, Yibing Song, Jue Wang, and Pengtao Xie. Not all patches
> are what you need: Expediting vision transformers via token reorganizations. In International
> Conference on Learning Representations, 2022.

---

### Official Review · Reviewer_8Ziy · 2022-11-03

**Confidence:** 4
**Correctness:** 3
**Technical Novelty And Significance:** 3
**Empirical Novelty And Significance:** 3
**Recommendation:** 8

**Clarity, Quality, Novelty And Reproducibility:**

Clarity: As I mentioned above, the paper is relatively clear in writing, with some minor issues. I think they are not major if one continues reading and does multiple passes, but would be much more friendly to readers who are confused that MAE does not have momentum encoders.

Novelty: I think the exploration on reducing the computation complexity with progressive downsampling is interesting and new. I have not seen it before.

Reproducibility: It is a concern here. The paper did not mention about the code/model release, and did not reveal important hyper-parameters (e.g., the momentum coefficient, the learning rate, batch size etc.). I don't think readers have enough information to easily reproduce the results in the paper.

**Strength And Weaknesses:**

Strengths:
- The goal of reducing the number of reconstructed patches and the idea of progressively discarding patches are interesting and new to me. Regardless of how significant the progress is made in terms of accuracy or training efficiency, the exploration is also revealing quite interesting properties of the original MAE.
- The paper is relatively well-written, and the illustrations are professional and clean.
- I particularly like that the paper is highly focused on the goal of reducing the number of reconstructed patches, and does extensive evaluations on the removal strategies. This gives a more complete picture of the proposed idea.

Weaknesses:
- My biggest concern is about the computation advantage over MAE when the encoder gets larger. Right now the analysis (especially on speed) is completely dependent on ViT-B -- this is where MAE has the least advantage due to the use of a fixed-sized decoder and the size ratio between the encoder and the decoder in this case is the smallest. I believe the speed gain may not be so healthy if one uses a larger encoder, as the momentum encoder requires an extra forward pass through the encoder.
- While I think the paper is relatively well-written, some of the paragraphs in the abstract and in the introductions are not so welcoming for readers. For example, before introducing that the method contains a momentum encoder, it is already talking about doing the removal using the momentum encoder.
- I think using the "crops" to compare methods in Table 1 is a bit misleading. Right now PCAE indeed only uses 1 crop from each image in each iteration, but it does *2* forward passes through the encoders (one base, one momentum), at least one plus a partial one. The computation complexity is not reflected in the number of crops, but rather in the number of forward passes and backward passes.

Questions:
- What if we just have another pre-trained model (potentially self-supervised) just do the judge about which patches are more important/salient? The model can even be small. The model can even be non-neural network based. For example, one can just compute a saliency map, and remove patches based on the average saliency map within each patch? Maybe the computation can be further saved without even using a momentum encoder?

**Summary Of The Paper:**

The paper presents an improvement for MAE pre-training, that instead of reconstructing all the patches, it attempts to reconstruct only a certain number of diverse/important patches. The selection for diverse/important patches is done through a momentum encoder -- it will discard patches that are closest to the mean (as it is empirically corresponding to the low-frequency, background patches), and only use the remaining patches for MAE decoder to predict. Such an idea is shown to perform well on ViT-B, with both faster training and (slightly) better results than MAE. The idea of progressively discarding patches is also attempted during the fine-tuning stage as well.

**Summary Of The Review:**

Overall I am leaning toward acceptance. The paper has the merit that explores further reducing the computation complexity by discarding redundant patches along the way, which no one has done before if I remember correctly. The writing and illustrations are relatively clear. The results are showing some improvement over the baseline. However, I do have concerns about the scalability of the approach (to larger models), and the reproducibility.

---

> ### Author Response · Authors · 2022-11-12
> **Response to Reviewer 8Ziy Part I**
>
> We appreciate your insightful comments and valuable suggestions, and we respond to your concerns in the following:
>
> - **Q1: The scalability of PCAE to larger models**
>
>     As we posted in the general response Q1, PCAE not only accelerates the throughput but also converges much faster compared with MAE. Besides, PCAE also gains robust to extremely high mask ratios where MAE fails. These merits enable PCAE to keep advantages over MAE in large models.  We run PCAE on ViT-L with 85% mask ratio and compare it with MAE in the following table (the performance of MAE refer to Figure 7 in MAE paper.) Note that the performance of MAE suffers serious degradation under 85% mask ratio (around 83.8 800ep according to Figure 5 in MAE paper), and thus we compare the performance of MAE under 75% mask ratio.
>
>     |Methods    | epoch      | mask ratio | Throughput | GPU day    | Acceleration ratio     | FT acc     |
>     |:------    | :---------:|:----------:|:----------:|:----------:|:----------------------:|:----------:|
>     |MAE        | 800        | 75%        | 196.9      | 64.4       | 1 $\times$             | 84.9       |
>     |PCAE       | 300        | 85%        | 311.4      | 14.7       | 4.4 $\times$           | **85.1**   |
>     |MAE        | 200        | 75%        | 196.9      | 16.1       | 1 $\times$             | 83.3       |
>     |PCAE       | 100        | 85%        | 311.4      | 4.9        | 3.3 $\times$           | **83.6**   |
>
>     The results above show that PCAE achieves comparable performance to MAE with only around 1/3 or 1/4 GPU days, which grounds the advantage of PCAE over MAE in large models.
>
>     As the complexity of vision transformers is quadratic to the number of tokens, PCAE benefits more for models using small patch size or high-resolution inputs (for more details please refer to our recent response to R2 Q2), and these two aspects are usually applied in large models. For example, the largest model applied in MAE paper is ViT-H/448 which applies patch size of 14 and 448 resolution. We measure the throughput of PCAE and MAE on ViT-H/448 and find that PCAE could accelerate 2.3 times throughput compared with MAE, which maintains the advantage of PCAE over MAE.
>
>     We further discuss the impact of the momentum encoder. We discard tokens during the forward of the momentum encoder, and thus its cost is largely reduced. Take ViT-H/448 as an example, we find the forward of the momentum encoder only takes around 18% overheads even with such a huge model. Therefore, the cost of the momentum encoder will not affect the acceleration of PCAE over large models too much. Besides, thanks to your valuable suggestion, we find PCAE works well when we replace the momentum encoder with a small pre-trained model, which enables PCAE to further reduce the cost from the extra forward.
>
> - **Q2: The writing of momentum encoder**
>
>     We are sorry for it and will fix it in later revisions.
>
> - **Q3: The crops in Table 1**
>
>     Thanks for this suggestion! We will replace the crop with the number of the forward pass in our revision.
>
> - **Q4: Importance identify via other pre-trained models**
>
>     Thanks for your suggestion! We use the DINO [1] pre-trained ViT-S model (as MAE does not provide pre-trained ViT-S models) to identify saliency tokens and compare it with PCAE. Specifically, we follow DINO and regard the attention score to the cls token as token saliency. We apply the progressive strategy on the pre-trained model, and identify redundant tokens according to the saliency or PCAE respectively. Then, we force the student branch to reconstruct the output of the pre-trained model. We term this setting as Saliency-token or PCAE-token respectively.
>     We run training for 100 epochs and report their fine-tuning performance in the following table.
>
>     |   Setting           | FT acc.     |
>     |:--------------:     |:-----------:|
>     |  PCAE               | 82.3        |
>     |  Saliency-pretrain  | 82.7        |
>     |  PCAE-pretrain      | 83.1        |
>
>     We compare Saliency-pretrain with PCAE-pretrain and find that the saliency based method is inferior to PCAE. It is caused by saliency based method performs worse in identifying important tokens at intermediate layers, especially at shallow layers (a recent study [2] also reports this issue). Therefore, saliency based methods is inferior to PCAE. Comparing PCAE and PCAE-pretrain, we find that the pre-trained model helps PCAE converge faster even though it is a relatively light-weight model. Therefore, it is possible to replace the momentum encoder with a small pre-trained model, which improves the efficiency of PCAE when deployed on large models. We think this is a valuable suggestion and also post it in the general response.

---

> > ### Author Response · Authors · 2022-11-16
> > **Response to Reviewer 8Ziy Part II**
> >
> >
> > - **Q5: Reproducibility**
> >
> >     We provide our implementation details in the appendix section B and feel sorry for you missing it. Specifically, we follow the implementation of MAE and apply its optimization hyper-parameters including learning rate, schedule, and optimizer for fair comparisons. We set fix momentum equal to 0.9999. We will also release our code and pre-trained models once this paper is accepted by ICLR2023 to ensure its reproducibility.
> >
> > We sincerely thank for your efforts in reviewing this paper, and we are looking forward to your response.
> >
> > [1] Caron, Mathilde, et al. "Emerging properties in self-supervised vision transformers." Proceedings of the IEEE/CVF International Conference on Computer Vision. 2021.
> >
> > [2] Xu, Yifan, et al. "Evo-vit: Slow-fast token evolution for dynamic vision transformer." Proceedings of the AAAI Conference on Artificial Intelligence. Vol. 36. No. 3. 2022.

---

### Official Review · Reviewer_u9Ph · 2022-11-03

**Confidence:** 4
**Correctness:** 3
**Technical Novelty And Significance:** 2
**Empirical Novelty And Significance:** 2
**Recommendation:** 6

**Clarity, Quality, Novelty And Reproducibility:**

There are some minor typos in the current draft, such as inconsistent use of parentheses for inline citations in Section 2.

The source code is not provided to help with the reproducibility.

I think the novelty of the paper is limited, refer to the main review.


**Strength And Weaknesses:**

**Main Review**

**Major Strengths**

The paper proposes a strategy to remove redundant masks from self-supervised training that is demonstrated to be quite effective in several tasks including object detection and segmentation.

It either accelerates throughput or improves memory performance in the reported experiments.

The experiments are comprehensive and help visualize the step-wise discarding method.

The ablation study is quite useful in choosing drop case, drop ratio, and drop stage which are three crucial hyper-parameters of the proposed method PCAE.


**Weaknesses**

Similarity measure in pixel space using mean statistics is fundamentally not so efficient, especially for images (Ref Equation 3). This is because natural images often lie on a low-dimensional manifold where a smaller distance in pixel space may not help identify images containing similar features. Similarity in feature space might be helpful, but not so much when mean is used rather than a learned metric as done in related works (Sec 2).

I am confused about how to choose the size of the patches. If I choose a large patch size, it may have high frequency components and a small patch size may have low frequency components. The algorithm may not consider patches to be redundant if there are too many high frequency components in it. On the other hand, there might be too many redundant patches if the size is small.
Also, how to choose the cut-off frequency to separate high and low frequency components while discarding redundant patches? What is the intuition behind choosing a threshold?

What is the optimal stopping point while discarding redundant patches and what is the criterion for choosing this optimal stopping point?

In Section 3.4, what is the intuition behind concatenating the average of dropped tokens with the retained ones? Is it just another heuristic? My main concern is that the patches are dropped based upon their similarity with the mean, which means the average of these dropped patches must be close to the mean. Then, what additional information does it provide when we append it to the retained ones?

In Appendix E, compared to RD, PCAE offers an improvement from 81.1 to 82.3, which does not seem significant when their computational complexity is considered. It is relatively easier to randomly drop patches than the proposed strategy as long as the performance drop is not so much.


**Summary Of The Paper:**

Self-supervised learning helps learn effective representation that is often useful in downstream tasks. Reconstruction of masked patches from visible ones is a crucial step in learning effective representation. However, reconstruction of the entire image seems redundant given most parts of a natural image are highly correlated. This leads to issues in computational and storage space complexity. This paper proposes PCAE to reconstruct some portions of the natural image by progressively dropping non-relevant patches based on mean similarity score. It uses the vision transformer to measure the leakage of information from discarded tokens to retained ones. The efficacy of the proposed strategy is demonstrated on large-scale pre-training datasets and several downstream tasks.


**Summary Of The Review:**

I do believe that the paper has some interesting ideas. However, there are so many statements as listed in the main review that seem adhoc. The reason or intuition for the proposed ideas is not well-motivated. Although the empirical results look good in terms of throughput and memory requirement, the methodology could be communicated more clearly.

After discussion with the authors, I am convinced that the paper has some major strengths that will benefit the community. Assuming that the authors will make the necessary changes in the final version, I am raising my score to borderline accept.

---

> ### Author Response · Authors · 2022-11-09
> **Response to Reviewer u9Ph**
>
> We sincerely thank for your efforts in reviewing this paper, and we respond to your concerns in the following:
>
> - **Q1: Mean statistics and learned metrics in related works**
>
>     We are confused about this "weakness" because **we use the mean statistics in the feature space (the $x$ in equation 3 indicates tokens)**. Experiments and visualizations show that it is sufficient to identify redundant tokens and leads to high-quality representations. Besides, we also show that previous methods are only accessible for the supervised setting (r.f. Sec. 2) and it is not comparable with PCAE under the unsupervised setting. Could you clarify your question?
>
> - **Q2: The selection of patch size**
>
>     The patch size is a hyper-parameter of the vision transformer. As it does not relate to our method, we apply the default configuration of MAE. We also confuse about the claim that "a large patch size may have high frequency components and a small patch size may have low frequency components". The frequency component only relates to the content of the patch rather than the patch size. Besides, PCAE measure the redundancy among tokens rather than inside tokens, which means tokens that have similar components will be discarded first whatever the frequency component the token contains. This is well verified by our visualization analysis in Section 3.3.2. Owe to the stage-wise strategy, PCAE discards different components at each stage, and finally retains tokens belonging to various objects. We are really confused about this question and strongly disagree that this is a weakness of this paper. Could you please clarify it?
>
> - **Q3： The optimal stopping point while discarding redundant tokens**
>
>     Obviously, the optimal point depends on the content of the images and various for each image. However, considering parallel training, we apply the same stopping point for all images. The optimal stopping point depends on the average redundancy of the pre-training dataset and needs empirical searches. We show extensive ablation studies in section 4.3, and the results show that retaining 12.5% patches leads to the best balance between performance and overheads.
>
> - **Q4: The average of dropped tokens in downstream tasks**
>
>     The dropped tokens are similar to the mean but not identical, and thus these tokens surely provide extra information for decision. Besides, retaining the fusion token allows the fusion token to be utilized by deeper layers, which alleviates useful information discarded at the shallower layers. This strategy is straightforward and well-motivated.
>
> - **Q5: The comparison between RD(random drop) and PCAE**
>
>     Initially, we want to clarify that 81.1 and 82.3 is a big differences in MIM methods. For example, referring to the MAE paper Table 3, models trained from scratch fall behind MAE pre-trained model by 1.3 points, but they show a much larger gap in COCO object detection task (MAE paper Table 4). Therefore, there is a nontrivial performance gap between RD and PCAE. Please justify your comments. Besides, considering PCAE has nearly the same complexity as RD, PCAE is significantly superior to RD, and it is unreasonable to choose a much less powerful method without any gains in efficiency.
>
> - **Q6: Reproducibility**
>
>     We will release our source code and pre-trained models once this paper is accepted by ICLR2023 to ensure its reproducibility.
>
> We are looking forward to your response.

---

> > ### Comment · Reviewer_u9Ph · 2022-11-12
> > **Thank you for the response**
> >
> > I thank the authors for the timely response. It partially answers my questions. I'll adjust my score based on the newly added discussion and the response to questions by the other reviewers. I still have the following concerns:
> >
> > In the main review, the weakness (1) clearly identifies the benefits of measuring similarity in the feature space. However, the benefits over a **learned metric** (as done in the previous works) should be explicitly discussed.
> >
> > Considering the diversity of natural scenes, a large patch is more likely to contain diverse features compared to a small patch. This leads to low similarity across tokens due to the diversity of features in large patches. On the other hand, if the patch size is small, then patches are more likely to have homogenous redundant features that can be discarded. In the later case, one needs to drop more patches compared to the former. Clearly, the performance of the proposed method (e.g. throughput and memory) depends on the patch size as well. Therefore, what happens to the performance of PCAE if the size of the vision transformer is tuned optimally? Does the improvement in throughput stay the same?
> >
> > I think the response needs to be properly quantified. `The dropped tokens are similar to the mean but not identical, and thus these tokens surely provide extra information for decision.` Again, here the authors measure similarity in mean statistics. What about the other moments? Is it similar in terms of distributional sense? Is it similar in terms of the entropy or information content? Extra information in what sense? Please quantify the information content or the mutual information to make a concrete statement?

---

> > > ### Author Response · Authors · 2022-11-14
> > > **Response to Reviewer u9Ph**
> > >
> > > We sincerely appreciate your timely responses.
> > >
> > > - **Q1: The benefits over a learned metric**
> > >
> > >     For identifying the redundant tokens, the learned metrics are usually based on supervisory signals to measure the relevance of each patch with respect to the target task like classification [1,2,3,4]. However, this is not suitable for self-supervised learning settings where supervisory signals are inaccessible and it is hard to automatically identify task-relevant tokens without explicit guidance. In contrast, PCAE removes similar tokens by computing the similarity of each token with the averaged token. In this way, PCAE does not rely on supervisory signals and thus is more suitable for self-supervised learning than these learned metrics, and we verify in experiments that such a strategy is effective in self-supervised learning.
> > >
> > >     For a more comprehensive comparison between PCAE and learned metrics, we adapt SOTA dynamic ViTs into self-supervised learning. Recent SOTA dynamic ViTs are dominated by saliency based methods [4,5]. These methods measure the importance of patches according to their attention to the class token. To mitigate the issue of lack of supervisory signals, we use a DINO [6] pre-trained model, which has been demonstrated to produce high quality saliency map for images, to identify token saliency. We apply the same implementation and discard tokens according to the produced saliency. The results show that PCAE performs better than this learned metric, for more details please refer to **General Response Q3**.
> > >
> > > - **Q2: The relation between patch size and performance**
> > >
> > >     Thanks for your explanation!
> > >
> > >     The improvement of PCAE on throughput comes from reducing the number of tokens. Thus, the relation between efficiency gains and the number of tokens can be given as:
> > >     $g\propto\frac{N^2 + \beta N}{(N \alpha)^2 + \beta N \alpha}=1/\alpha^2 - \frac{\beta(1-\alpha)}{N}$ where $N$ denote the number of tokens which depends on the patch size, $\alpha$ is the keep ratio, $\beta$ is the relative ratio of operations whose complexity is linear to the number of tokens. **This indicates that the benefits of PCAE increase as the number of tokens increases**.
> > >
> > >     For a given resolution, the quadratic of the number of tokens is inversely proportional to the patch size. Thus, the advantage of PCAE will be more significant when the patch size is smaller. Too big patch sizes would significantly deteriorate the performance and are not usually used in practice. For widely used patch size, i.e., 8, 14, 16 [6,7], the advantages of PCAE is obvious. We report the throughput improvement of PCAE with different patch sizes with a mask ratio of 85% (r.f. **General Response Q1**):
> > >
> > >     | Patch size/tokens  |  16/196  |  14/256  |  8/784   |
> > >     | :----------------- | :--: | :--: | :--: |
> > >     | Acceleration ratio | 2.25 | 2.40 | 3.55 |
> > >
> > >     To verify what PCAE discards under different patch sizes, we further compare their visualizations and post them in the **rebuttal revision appendix G**. The visualizations show that PCAE discards similar things under different patch sizes, which indicates that PCAE is robust to the patch size.
> > >
> > > - **Q3: Clarify the average token in the downstream task**
> > >
> > >     Denote the whole token sequence,  the discarded tokens, and the remained tokens as random variables $X$, $D$, $Z$, respectively, and the task-relevant variable (for example, the ground truth) as $Y$. In common practice, discarding tokens causes performance degradation, and thus we have $I(D, Y|Z)>0$. **Here $I(D,Y|Z)$ is what "extra information" means**. Since we discard tokens according to their similarity to the average token, most discarded tokens are similar to each other (refer to our visualization please). In this case, we represent these discarded tokens with their centroid, i.e, average. Our empirical results also show that this strategy slightly improves performance (+0.1%) in downstream tasks, which indicates that the appended token helps the decision.
> > >
> > >     Note that we choose the average token strategy simply following the token drop works in dynamic ViTs [4,8], which weighted average dropped tokens for better performance. In our experiment, it is a scalable setting, and we can choose to use it for slightly better accuracy (+0.1%) or not for slightly better throughput (+1%), without affecting the performance too much.
> > >
> > >     It is unclear to us how other moments can be used to identify redundant tokens. In my understanding, other moments are not in the same feature space as tokens. For example, the second order moment reflects the variance of each dimension, and its range of values along with practical meaning differs significantly from tokens. Thus, the distance between these moments and tokens cannot be used to identify redundant tokens.
> > >
> > > We are looking forward to your response.

---

> > > > ### Author Response · Authors · 2022-11-14
> > > > **Reference**
> > > >
> > > >
> > > > [1] Rao Y, Zhao W, Liu B, et al. Dynamicvit: Efficient vision transformers with dynamic token sparsification[J]. Advances in neural information processing systems, 2021, 34: 13937-13949.
> > > >
> > > > [2] Yin H, Vahdat A, Alvarez J M, et al. A-ViT: Adaptive Tokens for Efficient Vision Transformer[C]//Proceedings of the IEEE/CVF Conference on Computer Vision and Pattern Recognition. 2022: 10809-10818.
> > > >
> > > > [3] Meng L, Li H, Chen B C, et al. AdaViT: Adaptive Vision Transformers for Efficient Image Recognition[C]//Proceedings of the IEEE/CVF Conference on Computer Vision and Pattern Recognition. 2022: 12309-12318.
> > > >
> > > > [4] Youwei Liang, Chongjian Ge, Zhan Tong, Yibing Song, Jue Wang, and Pengtao Xie. Not all patches
> > > > are what you need: Expediting vision transformers via token reorganizations. In International
> > > > Conference on Learning Representations, 2022.
> > > >
> > > > [5] Xu Y, Zhang Z, Zhang M, et al. Evo-vit: Slow-fast token evolution for dynamic vision transformer[C]//Proceedings of the AAAI Conference on Artificial Intelligence. 2022, 36(3): 2964-2972.
> > > >
> > > > [6] Caron M, Touvron H, Misra I, et al. Emerging properties in self-supervised vision transformers[C]//Proceedings of the IEEE/CVF International Conference on Computer Vision. 2021: 9650-9660.
> > > >
> > > > [7] He K, Chen X, Xie S, et al. Masked autoencoders are scalable vision learners[C]//Proceedings of the IEEE/CVF Conference on Computer Vision and Pattern Recognition. 2022: 16000-16009.
> > > >
> > > > [8] Kong, Zhenglun, et al. "SPViT: Enabling Faster Vision Transformers via Soft Token Pruning." arXiv preprint arXiv:2112.13890 (2021).

---

> > > > > ### Comment · Reviewer_u9Ph · 2022-11-15
> > > > > **Thank you for the response**
> > > > >
> > > > > I thank the authors for including the new results and discussion in the updated draft. Having considered the suggestions and corresponding responses in the revised version, I am leaning towards raising my score to borderline accept.

---

> > > > > > ### Author Response · Authors · 2022-11-15
> > > > > > **Thanks for Your Response**
> > > > > >
> > > > > > We appreciate your kind response. Many thanks for your efforts and recommendations which make our paper better.

---

### Official Review · Reviewer_B38C · 2022-11-04

**Confidence:** 4
**Clarity, Quality, Novelty And Reproducibility:** Refer to my above comments.
**Correctness:** 3
**Technical Novelty And Significance:** 2
**Empirical Novelty And Significance:** 2
**Recommendation:** 5

**Strength And Weaknesses:**

Pros:
1. The proposed method is simple and easy to follow.
2. Experimental results demonstrate the method can significantly improve the training efficiency of MAE in terms of both throughput and memory cost, while still retaining the performance.

Cons
1. The motivation for improving efficiency by reducing reconstructed tokens is questionable. The authors claim that the decoder of MAE takes up most training costs, which however only holds when the decoder is deep. Actually, as pointed out in the original MAE paper, applying a lightweight decoder would not significantly affect its performance. In this case, the decoder gets efficient so that it is unnecessary to use a small set of mask tokens. Further, in another SSL method SIMMIM, the decoder is even more lightweight than in MAE, rendering the applicability of PCAE limited.

2. Several things are lacking in the analysis. More work needs to be done for verification.
- In Sec. 3.1.1, the authors illustrate the diversity of tokens. How is this diversity related to the representation power of the model? Why does PCAE has similar top1 results on Imagenet1k as  MAE even though its tokens are more diversified?
- If other methods, like using saliency to select tokens, can achieve comparable results with the proposed method.
- It is unclear how to utilize the average of abandoned tokens in downstream task.
- The sensitivity of token dropping ratio on the performance of downstream tasks.

3. Overall, the performance of PCAE is marginally above that of MAE as shown in Table 1 and Table 2, suggesting the learned representations in this way have little to no improvement over the baseline.

**Summary Of The Paper:**

In this paper, the authors propose a method to accelerate the training of MAE by reducing the number of tokens to reconstruct. Specifically, the authors use a teacher-student training pipeline where the teacher operates on mask tokens to progressively identify those redundant tokens among them, finally providing a compact token set as the reconstruction target of the student network. A simple ranking method is proposed for scoring the importance of tokens by calculating the similarity of each token with the mean value.

**Summary Of The Review:**

Refer to my above comments.

---

> ### Author Response · Authors · 2022-11-09
> **Response to Reviewer B38C**
>
> We appreciate your insightful comments and valuable suggestions, and we respond to your concerns in the following:
>
> - **Q1: The motivation of PCAE by reducing reconstructed tokens.**
>
>     As we clarified in the general response, MAE relies on deep decoders for better performance in downstream tasks like linear probe, detection, and segmentation, and thus **8-layer decoders are the default configuration of the original MAE paper**. Besides, PCAE also works in extremely high mask ratios where MAE usually fails, which further helps PCAE accelerate MAE.
>
>     As we posted in the general response, **PCAE also can accelerate SimMIM-like methods by reducing masked tokens in the encoder**. Since the encoder is more complicated, SimMIM benefits more from PCAE compared with MAE. Specifically, PCAE accelerates SimMIM at most 2.27 times without compromising performance (PCAE-SimMIM 82.3 vs SimMIM 82.1 100 epoch).
>
> - **Q2: Things need verification**
>
>     - **The diversity of tokens w.r.t. performance**
>
>         We illustrate in Figure 3 to validate the property of PCAE by recovering diverse tokens, this is because we only reconstruct the key patches and remove those redundant ones, this is the core principle for acceleration. **Note that the core contribution of PCAE is for accelerating MAE pretraining, and we only recover some diverse tokens to achieve this goal, not diverse tokens for better performance**.
>
>     - **Alternative methods like saliency maps to select tokens.**
>
>         Thanks for the suggestions. We agree that the saliency of tokens can remove redundant tokens from reconstruction targets and relax the reconstruction task.
>         However, the saliency map is hard to obtain in self-supervised learning due to the lack of supervisory signals, and previous dynamic token selection methods are only accessible for the supervised setting [1], [2]. As far as we know, we are the first to consider patch relevance and drop those redundant ones in self-supervised settings. Here, we further try two ways to obtain sailency of tokens and post their results and analysis to them in the **General response Q3**. The results show that PCAE performs better than saliency based methods under the unsupervised setting.
>
>         Besides, we also want to clarify the differences between the proposed strategy and token saliency. PCAE are motivated by reducing the gap between the original targets and the compressed ones, and thus removes redundant tokens whether they belong to the foreground or background while saliency based methods only expect to keep the foreground. This property is well verified by the visualization in Figure 2 (for more details please refer to our analysis in section 3.3.2). Therefore, PCAE is more similar to redundancy reduction methods rather than saliency based methods. We think other efficient redundancy reduction methods can achieve comparable performance to PCAE. However, the metric of similarity to mean applied in PCAE is very efficient and leads to high-quality representations, and thus we apply it across our experiments.
>
>     - **How to utilize the average of abandoned tokens in downstream tasks.**
>
>         We simply append the average of abandoned tokens after retained tokens and keep it in later blocks as this way alleviates information loss caused by token reduction. We find this strategy sightly improves fine-tuning performance (+0.1%).
>
>     - **The sensitivity of token dropping ratio in downstream tasks**
>
>         We show the relation between the drop ratio and fine-tuning performance in Table 6. The results show that the performance drop is within 0.6% when retaining 34.3% tokens (0.7^3), and performance suffers serious degradation when the drop ratio exceeds 0.3.
>
> - **Q3: The performance improvement of PCAE is marginal**
>
>     We clarify that the core contribution of PCAE is acceleration rather than performance gain. Referring to Table 1, **PCAE achieves comparable performance to MAE with around 1/8 overheads**. The performance and overheads are two sides of the coin, and it is unreasonable to talk about performance while neglecting its cost.
>
> We sincerely thank for your efforts in reviewing this paper, and we are looking forward to your response.
>
> [1] Yongming Rao, Wenliang Zhao, Benlin Liu, Jiwen Lu, Jie Zhou, and Cho-Jui Hsieh. Dynamicvit:
> Efficient vision transformers with dynamic token sparsification. Advances in neural information
> processing systems, 34:13937–13949, 2021.
>
> [2] Youwei Liang, Chongjian Ge, Zhan Tong, Yibing Song, Jue Wang, and Pengtao Xie. Not all patches
> are what you need: Expediting vision transformers via token reorganizations. In International
> Conference on Learning Representations, 2022.

---

> ### Author Response · Authors · 2022-11-17
> **Thanks and Looking Forward to Your Reply**
>
> Dear Reviewer,
>
> we would appreciate your valuable suggestions which helps us a lot in improving our paper. We wonder if there is any additional question and would like to answer at any time. If our response has cleared your concerns, could you please consider re-evaluate our paper?
>
> Many thanks for your comments again.

---

### Author Response · Authors · 2022-11-09
**General Response part I**

We sincerely appreciate all reviewers for their efforts in reviewing this paper. We first give general responses for some common issues.

**Q1: The motivation and advantages of PCAE by reducing reconstructed tokens.**

- The core idea of PCAE is to reduce the redundancy in reconstruction targets in MAE for acceleration. We find that the naive MAE is training inefficiently due to the redundant reconstruction task, and it deteriorates pre-training in the following two aspects:

    - First, recovering all masked patches is optimization difficult as the reconstruction goal may focus on recovering some trivial details that contribute little to semantic understanding, this is why MAE needs 1600 epochs training. Besides, this is also computation costive, especially for reconstruction with deep decoders. Note that the relatively deep decoder is necessary for better performance in downstream tasks like linear probing (referring to Table 1 (a) in the MAE paper, the linear accuracy drops by 8 points from 8-layer decoder to 1-layer decoder), and that is why **the default setting of MAE uses 8-layer decoder**. A recent study [1] also points out that the deep decoder is necessary for downstream tasks like detection and segmentation;

    - Second, the heavy reconstruction task forces MAE to keep a considerable ratio of visible patches in the encoder, and thus MAE suffers serious performance degradation once its mask ratio exceeds 75%, which limits its further acceleration by masking more tokens.

    In contrast, PCAE removes redundant reconstruction targets and largely relaxes the reconstruction task.
    As a result, **the acceleration of PACE comes from two aspects, one for adding throughout via dropping more tokens, and the other for faster convergence via free of reconstruction redundant patches**. In practice, PACE not only accelerates throughput 1.5-1.9x times in training but also converges faster as it avoids focusing on recovering redundant patches. This is verified in by extensive experiments. As shown in Table 1, PCAE with only 300 epochs achieves comparable performance (83.6 vs 83.6) with MAE that even training for 1600 epochs, which is also a significant acceleration to MAE. Overall, **the total overheads of PCAE are 1/8 of MAE (10.1 GPU days vs 82.1 GPU days) when reaching comparable performance, and we think it is not a marginal improvement**. Benefiting from the relaxed reconstruction task, PCAE also works in extremely high mask ratios that the naive MAE usually fails.

    |Mask ratio | 75%        | 85%        | 90%      |
    |:------    | :---------:|:----------:|:--------:|
    |FT acc.    | 83.6       | 83.6       | 83.5     |
    |COCO det.  | 47.6       | 47.7       | 47.4     |

    As presented in the above table, it nearly does not affect the performance of PCAE even if we mask 90% of tokens of input images while MAE suffers serious performance degradation once its mask ratio exceeds 75% (MAE drops 1.9 points when raising the mask ratios from 75% to 90%, and for more details please refer to Figure 5 in MAE paper). PCAE only reconstructs around 10% of tokens while inputting 25% of tokens, which makes the reconstruction task relatively simple. Therefore, the encoder could tolerate higher mask ratios on the input. This merit enables PCAE further accelerates MIM methods by applying higher mask ratios on the input images, and we find it even could accelerate MAE 2.7 times when we apply the 90% mask ratio. This interesting property reveals the fact that the overhead in both the input end and output end could be relaxed simultaneously for efficient training, which could help future works in designing large-scale pre-training.

[1] Chen, Y. et al. (2022). SdAE: Self-distillated Masked Autoencoder. ECCV 2022. Lecture Notes in Computer Science, vol 13690. Springer, Cham. https://doi.org/10.1007/978-3-031-20056-4_7

**Q2. The acceleration for methods using light-weight decoders**

We clarify that **PCAE accelerates MIM methods by relaxing the reconstruction task rather than the "decoder"**, and thus PCAE still works for MIM methods using light-weight decoders like SimMIM. SimMiM alike methods usually keep the masked tokens in the encoder and recover masked patches through the encoder. PCAE makes them only recover important patches and thus accelerates them by reducing the number of masked tokens in the encoder. Since the encoder is more complicated, these methods are more sensitive to the redundant reconstruction tasks and benefit more from PCAE. As we show in Appendix D, we adapt PCAE into SimMIM and accelerate it at most 2.27 times without compromising performance (PCAE-SimMIM 82.3 vs SimMIM 82.1 100 epoch).

---

> ### Author Response · Authors · 2022-11-12
> **General Response part II**
>
>
> **Q3: Other alternative options (saliency?) for removing redundant tokens**
>
> We consider two settings here. In the first setting, we measure the token importance with the average attention from other tokens. Specifically, we average the attention matrix of different heads in the self-attention block and calculate the mean of each column to get the importance of each token. Then we discard less important tokens. We follow the same implementation and settings with PCAE. We term this setting as Token Importance. In the second setting, we replace the momentum encoder of PCAE with a DINO [1] pre-trained ViT-S model, and identify redundant tokens according to the saliency produced by pre-trained model or PCAE respectively for comparison. We term this setting as Saliency-Pretrain or PCAE-Pretrain. We run these settings for 100 epochs and report their fine-tuning accuracy in the following table.
>
> |   Setting           | FT acc.     |
> |:--------------:     |:-----------:|
> |  PCAE               | 82.3        |
> |  Token-Importance   | 80.7        |
> |  Saliency-Pretrain  | 82.7        |
> |  PCAE-Pretrain      | 83.1        |
>
> The results shows that Token-Importance performs much worse compared with PCAE (80.7 vs 82.3). We owe it to the lack of supervisory signals. The network is hard to identify important tokens at the beginning stage where parameters are randomly initialized, and the worse beginning propagates errors to later phases, resulting in serious performance degradation ultimately. We also find that PCAE-Pretrain performs better than Saliency-Pretrain. It is caused by the diffculty in identifying redundant tokens at intermediate layers, especially at shallow layers for the saliency based method (a recent study [2] also reports this issue), and thus it may discard important tokens and causes performance degradation.
>
> [1] Caron, Mathilde, et al. "Emerging properties in self-supervised vision transformers." Proceedings of the IEEE/CVF International Conference on Computer Vision. 2021.
>
> [2] Xu, Yifan, et al. "Evo-vit: Slow-fast token evolution for dynamic vision transformer." Proceedings of the AAAI Conference on Artificial Intelligence. Vol. 36. No. 3. 2022.

---

### Author Response · Authors · 2022-11-15
**Modifications to the revision**


Dear Chairs and Reviewers,

We sincerely appreciate your efforts on review works. We have analyzed these review comments sentence by sentence carefully and revised our paper following the suggestions. All the revised contents in the revision are colored in blue and the main changes are summarized as follows:

Introduction:
1. We add the discussion to SimMIM alike methods to show that PCAE also can be adapted into these methods.
2. We clarify the description of removing part of the reconstruction targets.
3. We modify the description of the momentum encoder to make it easier to follow.

Related work:
1. We add [1] to our related works and discuss it in this section.

Experiments:
1. We replace the "crops" with "forwards" in Table 1.

Method:
1. We correct equation (1) by removing the error parenthesis.
2. We correct the description to $\omega$.
3. We replace $N$ with $N_i$ to indicate its value is various for different layers.
4. We denote the redundant token identification function as $T(\cdot;\cdot)$ instead of $D(\cdot;\cdot)$ to avoid ambiguous.
5. We correct the summation over $x^j$ in equation (3).

Appendix:
1. We add more details of PCAE-SimMIM in Appendix D.
2. We add the comparison to other strategies in Appendix E.
3. We add the visualizations under different patch sizes in Appendix G.
4. We add ablation studies of mask ratio, decoder depth, and architectures in appendix H.

We sincerely thank you for your insightful comments and valuable suggestions which help us improve this paper. We are happy to answer any additional questions at any time.

[1] Zeng, Wang, et al. "Not All Tokens Are Equal: Human-centric Visual Analysis via Token Clustering Transformer." Proceedings of the IEEE/CVF Conference on Computer Vision and Pattern Recognition. 2022.

---

### Decision · Program_Chairs · 2023-01-20

**Decision:**

Accept: poster

**Justification For Why Not Higher Score:**

The paper's experimental results and analyses could be improved.

**Justification For Why Not Lower Score:**

The overall quality is good, and five out of six reviewers liked the paper.

**Metareview: Summary, Strengths And Weaknesses:**

Six experts reviewed the paper. Four reviewers recommended it "marginally above the acceptance threshold" with high confidence, one reviewer labeled it as a good paper, and only reviewer B38C placed it "marginally below the acceptance threshold". Though reviewer B38C raised questions about the paper's experimental results and analyses, AC found the authors' rebuttal convincing. Hence, the decision is to recommend the paper for acceptance. The authors are encouraged to make the necessary changes to the paper to the best of their ability following the reviewers' comments.

**Note From Pc:**

if the above contains the word "oral" or "spotlight" please see: "oral" presentation means -> notable-top-5% and "spotlight" means -> notable-top-25%. As stated in our emails, we are disassociating presentation type from AC recommendations